# Small-to-Large Generalization: Data Influences Models Consistently Across Scale

**Alaa Khaddaj**
alaakh@mit.edu
MIT

**Logan Engstrom**
engstrom@mit.edu
MIT

**Aleksander Mądry**
madry@mit.edu
MIT

## Abstract

Choice of training data distribution greatly influences model behavior. Yet, in large-scale settings, precisely characterizing *how* changes in training data affects predictions is often difficult due to model training costs. Current practice is to instead extrapolate from scaled down, inexpensive-to-train proxy models. However, changes in data do not influence smaller and larger models identically. Therefore, understanding how choice of data affects large-scale models raises the question: how does training data distribution influence model behavior across compute scale? We find that small- and large-scale language model predictions (generally) *do* highly correlate across choice of training data. Equipped with these findings, we characterize how proxy scale affects effectiveness in two downstream proxy model applications: data attribution and dataset selection.

## 1 Introduction

When training large-scale models, we often want to understand how changing the training data distribution influences model behavior. For example, we may ask: does adding a data source improve accuracy? Does removing a data source increase toxicity? However, answering such questions is difficult in practice as the cost of model training makes training on each data distribution (and comparing the resulting models) infeasible.

To overcome compute costs, current practice is to approximate large-scale model behavior with that of small-scale models. In this approach, one (a) calculates how a given change in data distribution changes small-scale (low-cost) models (e.g., by retraining small models with and without the change), then (b) extrapolates the corresponding influence for large-scale model predictions using insights from (a). Indeed, small-scale *proxy models* are a standard primitive in methods for dataset selection and cleaning (Engstrom et al., 2024; MosaicML, 2023a; Xie et al., 2023a; Chen et al., 2023).

Nevertheless, there is yet no precise characterization of when proxy models are effective. After all, model behavior often changes across scale (Wei et al., 2022b); thus, changes in data may not influence small- and large-scale models identically. Understanding how training data changes large-scale model behavior therefore hinges on the question: how does training data influence model behavior across compute scale?

**Contributions.** After training language models (LMs) on a diverse set of training data distributions at different scales, we find that the answer is nuanced. On one hand, choice of training data distribution generally affects model predictions (very) similarly along compute scale (down to 175× smaller than the large-scale reference model, cf. Figure 1). Indeed, such a relationship even holds when proxy models are so small that their predictions are as accurate as *randomly guessing*.

On the other hand, however, our results also indicate that proxy models are not a panacea: we identify setups for which proxy model predictions do not correlate well with larger models. We find that only (very) small proxy models—those 370× smaller than the large-model class of interest—tend to predict larger-scale model behavior poorly.

Equipped with these findings, we then characterize the relationship between proxy model scale and performance in two downstream proxy model applications: data attribution (in vision settings) and dataset selection (in an LM setting) for large models. In both applications, we find that orders-of-magnitude smaller proxy-models can be as effective as using the original, larger-scale model of interest directly—but also that there is a clear trade-off between performance and proxy-model size at the smallest scales we study.

## 2 DATA INFLUENCE ACROSS SCALE

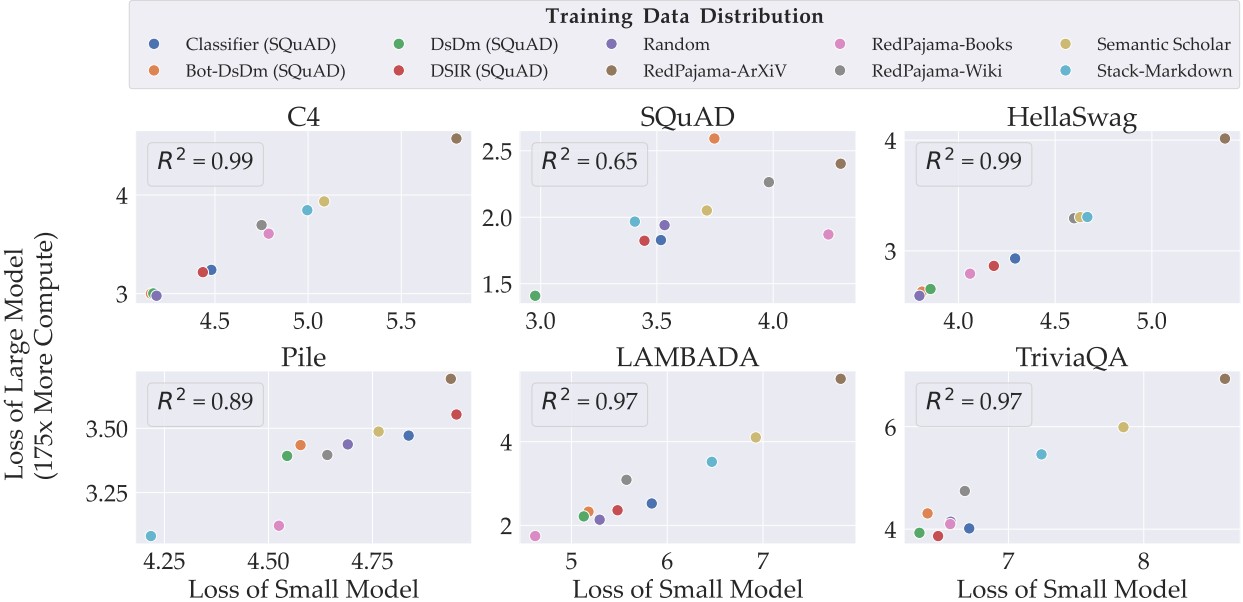

**Figure 1:** Proxy-model test loss highly correlates with large-model test loss across choice of training data distribution, even across a large gap in scale. Above, we plot the losses of a small-scale proxy (57M parameters) compared to that of the reference model (760M parameters). Here, the small scale model trains with 175× less compare than the reference model. Each column represents model loss on a different test distribution, ranging from LM benchmarks (SQuAD/HellaSwag) to pretraining data distributions (the Pile).

We seek to characterize how choice of training data influences model behavior across compute scale (i.e., the amount of compute used to train a model). To do so, we compare how changes in training data distribution affect large-scale model predictions compared to those of small-scale proxy models trained on the same data distributions. Correlating these differences across a diverse set of training data distributions, we find that training data generally influences model predictions similarly across scale—but that the degree of correlation depends on both the exact choice of test distribution and proxy model scale. In what follows, we first describe our experimental setup, then detail results (see Appendix B for additional details).

### 2.1 SETUP

We study how changes in data distribution affect the behavior of small *proxy* models compared to the behavior of a larger *reference* model class. We select 760M parameter language models as the reference model class (the largest setting that we can study in our available, academic-level compute budget). Our proxy models range in size from 40M parameters to 760M parameters, with each model training on a number of tokens determined by Chinchilla-optimal token-to-parameter ratios (Kaplan et al., 2020). In relative terms, these model train with down to 370× less compute than the reference model despite only having (at most) 19× fewer parameters (as they are trained with chinchilla-optimal token-to-parameter ratios).

We measure how model behavior changes across 10 separate training distributions: 6 *data-sources* (i.e., sampled from a single data source like Wikipedia (Foundation, 2022)) and 4 *selection-induced* distributions (i.e., data selected with one of three dataset selection methods: DSDM (Engstrom et al., 2024), DSIR (Xie et al., 2023b) and Classifier-based approach (Brown et al., 2020) using various target tasks). After training (separate) models on each of these training datasets, we compare the resulting model behavior (losses) on 6 test datasets: C4 (Raffel et al., 2020), the Pile (Gao et al., 2020), SQuAD (Rajpurkar et al., 2016), LAMBADA (Paperno et al., 2016), HellaSwag (Zellers et al., 2018) and TriviaQA (Joshi et al., 2017).

## 2.2 RESULTS

At a high level, we find that changes in training data distribution (generally) affect small- and large-scale model predictions similarly—even when the small proxy model is trained with much less compute than the large reference model in relative terms. We use the following basic primitive to study the effect of training data distribution: given downstream task, we measure the correlation of small- and large-scale losses across training data distributions. To obtain these results, we train small- and large-scale models on *each* training data distribution (one for each scale of model), and record the empirical loss of each of these models on downstream tasks.

We begin by studying the behavior of a single proxy model scale: 57M parameter proxy models. We relate in Figure 1 the losses of 57M parameter proxy models to those of the reference model class across different training data distributions, while varying (in each panel) the choice of downstream task. These proxy model losses (generally) highly correlate with those of large-scale models across training dataset, implying that choice of training dataset similarly changes both 57M and large (760M) model predictions—despite the proxy models training with 175× less compute.

To further study the role of proxy model scale, we relate in Figure 2 proxy model scale with the correlation between proxy and reference model predictions. We find that, as in the case of the 57M proxy model, losses are highly correlated. In general, losses are more correlated for proxy models that are closer in scale to the reference model.

However, our results also indicate that proxy models are not *always* reliable: the correlation between reference and proxy model predictions is highly dependent on (a) the gap in scale between the proxy and reference models (much smaller proxies are more mismatched) and (b) the exact choice of downstream task (proxy predictions are less correlated with reference model predictions on specific test distributions). For example, consider the smallest proxy model in Figure 2 (40M models, which use 370× less compute than the large model of interest). This class of model is highly correlated with the reference model on all the downstream tasks except two: SQuAD and TriviaQA (cf. Figure 2 for a detailed view).

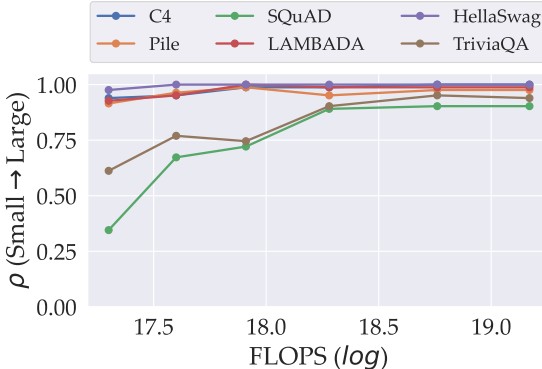

**Figure 2:** Correlation between large- and small-scale model predictions is consistently high, even across large gaps (orders of magnitude) in training compute scale. We plot small- to large-scale correlation against small-scale proxy model compute. There is also large variation across choice of test set: correlation is consistently high on four of six tasks, while losses on SQuAD and TriviaQA correlate less.

## 2.3 INTRIGUING PROPERTIES OF PROXY MODELS

We observe two additional properties of the relationship between proxy and reference models.

**Proxy models are effective regardless of accuracy.** We find that proxy model predictions for a given task can highly correlate with those of large-scale reference models *even* when the proxy models predict near the level of *random guessing* on that task. Indeed, relating proxy model accuracy against correlation with reference model predictions in Figure 3, we find that in two tasks—HellaSwag and COPA—small-scale proxy models achieve random-guessing level (or worse) accuracy while still highly correlating with large-scale models.

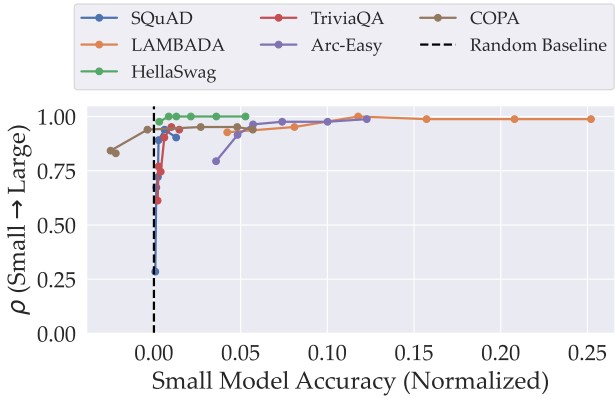

**Figure 3:** Proxy models can be highly predictive of large-scale model predictions even when predicting as well as randomly on a given test set. We plot small- to large-scale loss correlation against small-scale proxy model accuracy on the given task, normalized to show improvement over outputting a random guess (in absolute accuracy). On a number of test sets, proxy models perform no better than random guessing, but still highly correlate with the reference model (which always achieves significantly better than random guessing).

**Proxy models are (often) effective at a per-sample level.** We have thus far only studied the relationship between *average* losses achieved by proxy and reference models on each test task. To better characterize when proxy models match the reference model, we inspect similarity between small- and large-scale model predictions on *individual samples*—for individual test samples—in Figure 4. Our results indicate that proxy model predictions on individual samples can highly correlate with those of large models, depending on the choice of test dataset. On a population view, however, the picture is more nuanced: while proxy model predictions highly correlate with reference model predictions on the great majority of HellaSwag samples, they do not correlate as well on SQuAD samples (cf. Figure 5).

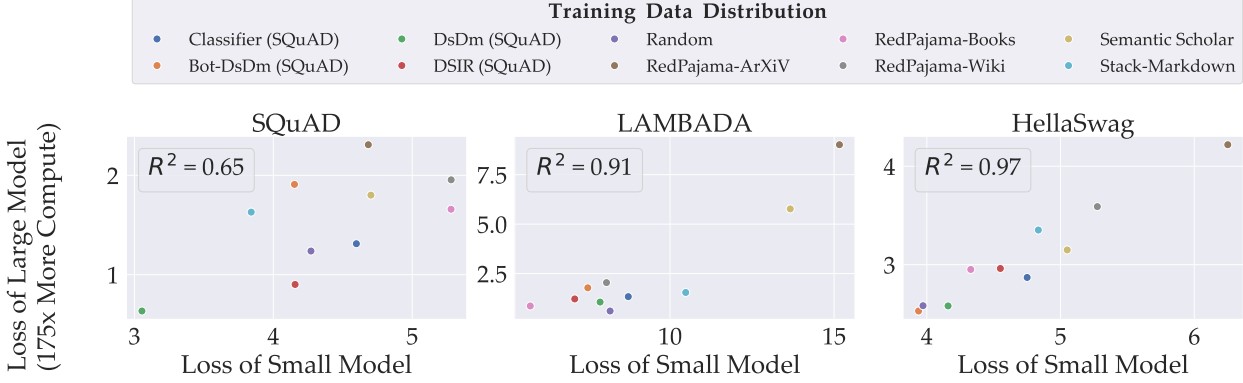

**Figure 4:** Proxy model predictions can highly correlate with those of the reference model on *individual* test samples. We visualize loss on individual samples for each scale model across varying training datasets. The proxy model here is 57M parameters, training with around 175× the compute of the 760M reference model. See a distributional plot (showing the correlation across *all* samples on each test set) in Figure 5.

## 3 PROXY MODELS IN DOWNSTREAM APPLICATIONS

Proxy model predictions generally highly correlate with reference model predictions across training distribution choice. However, at small proxy model scales this relationship can break down, suggesting that there is a fundamental trade-off between proxy compute scale and effectiveness. To understand how the proxy scale affects the utility of proxy models in downstream tasks, we characterize the role of proxy model scale in two downstream applications: attributing training data and selecting training data.

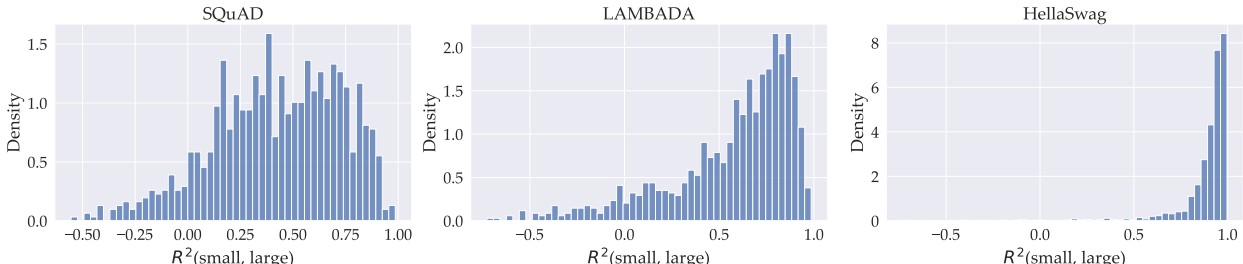

**Figure 5:** The correlation between large- and small-scale model losses on individual samples is highly dependent on the test distribution. We show a histogram of the correlation between large model and proxy model predictions on individual test samples for the test distribution in each column. We plot the coefficient of determination ($R^2$) between the losses of the small and large models on all examples in the downstream task.

### 3.1 ATTRIBUTING TRAINING DATA WITH PROXY MODELS

Data attribution methods analyze model behavior in terms of the training data (Koh & Liang, 2017; Ilyas et al., 2022). While these methods are helpful in tasks like dataset selection (Engstrom et al., 2024) and model debugging (Ilyas et al., 2022), they also tend to require compute that scales with the model size and the training dataset size. This requirement often makes data attribution prohibitively expensive in large-scale settings (Koh & Liang, 2017; Schioppa et al., 2022; Grosse et al., 2023). To make data attribution feasible at this scale, common practice is to instead attribute for a smaller proxy model, then use the result to attribute for the original model of interest (Engstrom et al., 2024).

#### 3.1.1 PRELIMINARIES

We start by defining data attribution within the datamodeling framework (Ilyas et al., 2022). Consider a training dataset $S = \{(x_1, y_1), \ldots, (x_n, y_n)\}$ of $n$ input-label pairs, and let $\theta(D)$ be the parameters of a classifier trained on subset $D$ of $S$. Then, given a sample $z = (x, y)$, let $f(z; \theta(D))$ be the loss of the classifier on $z$ after training on subset $D$ of the training set.

A *datamodel* for heldout sample $z$ is a simple (learned) function that estimates the final model loss on $z$ as a function of the subset $D$ used to train the model. For convenience, this is the function

$$\hat{f}_z(D) \approx f(z; \theta(D)),$$

which maps choice of training dataset to the loss of the resulting model on $z$. Intuitively, a datamodel $\hat{f}_z$ should accurately predict model loss after training on any given train subset $D$.

Previous work has found that the loss $f(z; \theta(D))$ can be approximated by *linear* datamodels, or datamodels that parameterize each training datapoint as contributing a fixed amount to the loss when included in the training dataset. That is, we can approximate the model loss reliably using the linear datamodel $\hat{f}_z$ parameterized as:

$$\hat{f}_z(D) := \sum_{i \in D} \tau(z)_i, \tag{1}$$

where $\tau(z)_i$ is a weight representing the "importance" of training example $i$ on predicting the heldout sample $z$ correctly.

**Estimating datamodel weights.** Families of approaches for estimating datamodel weights range from influence functions (Koh & Liang, 2017; Grosse et al., 2023) to resampling estimators (Feldman, 2019; Ilyas et al., 2022). In this work, we estimate datamodels using an influence function-based method called TRAK (Park et al., 2023). Briefly: TRAK estimates datamodel weights by (a) linearizing (trained) model output with respect to the model weights and then (b) calculating influences for this linearization (Koh & Liang, 2017). See Appendix A for full details and setup.

**Evaluating datamodels.** We evaluate datamodels with the Linear Datamodeling Score, or LDS (Ilyas et al., 2022; Park et al., 2023), a standard approach for evaluating data attribution methods (Bae et al., 2024; Zheng

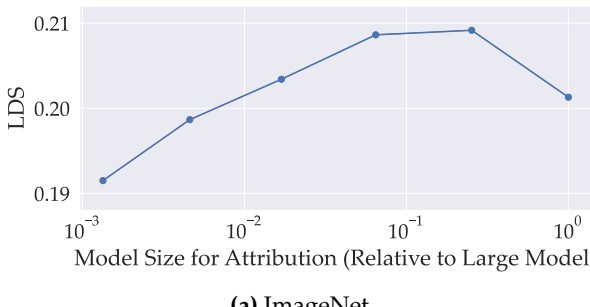

(a) ImageNet

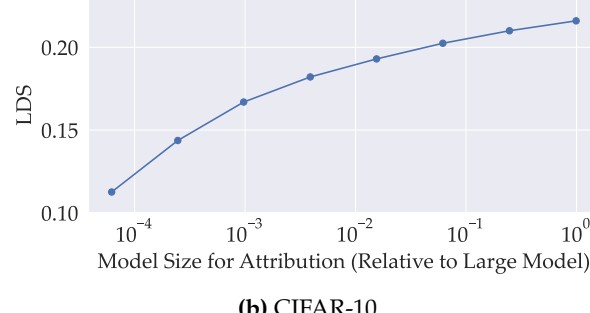

(b) CIFAR-10

**Figure 6:** In both plots, the *x*-axis represents the amount of compute required to get the attribution scores of a given model, compared to the large model, and the *y*-axis represents how well the attribution scores of a given model size can predict the output of the largest model on **(a)** CIFAR-10 and **(b)** CIFAR-100 respectively (Krizhevsky, 2009) (see Section 3 for details on the metric).

et al., 2024; Choe et al., 2024; Lin et al., 2024; Georgiev et al., 2023; Deng et al., 2024). For a heldout sample $z$, LDS measures the correlation between datamodel prediction of model loss and the actual model loss across $m$ randomly sampled training subsets $D_i$ (e.g., a common choice is to randomly choose fixed-size subsets of the training set). Specifically, the LDS for our linear datamodels is exactly the (Spearman) correlation:

$$LDS(\tau(z), z) := \rho_{\text{spearman}}\big(\underbrace{f(z; \theta(D_i)) : i \in [m]}_{\text{actual model loss}}, \quad \underbrace{\sum_{k \in D_i} \tau(z)_k : i \in [m]}_{\text{datamodel-predicted loss}}\big). \tag{2}$$

Intuitively, a datamodel that perfectly captures model loss would have an LDS of 1, and a datamodel that does not correlate with the model loss would have an LDS of 0. In this work, we measure the expected LDS over a given test distribution (by averaging LDS over test samples).

### 3.1.2 EXPERIMENTAL RESULTS

We study how well datamodels computed from smaller proxy models approximate the actual loss of the reference model in two supervised computer vision settings: ImageNet-1k (Russakovsky et al., 2015) and CIFAR-10 (Krizhevsky, 2009).

**Setup.** We estimate datamodels for ResNets (He et al., 2015) across a variety of model widths (ImageNet: the largest model class has a width $10^4$ times larger than the smallest; in CIFAR-10 this relative range is $10^5$). We then evaluate these datamodels by measuring the LDS with respect to the predictions of the *largest model class* (a $10^8$ parameter ResNet for ImageNet and $10^9$ for CIFAR-10). For additional details and results, see Appendix C.1.

**Results.** Small proxy models yield datamodel estimates that are similar in effectiveness to those calculated with the actual, large-scale model reference model. Relating proxy model size to LDS in Figure 6 (left) in the ImageNet setting, we find that LDS decreases in relative terms by (at most) 10% (from 0.21 to 0.19) across *all* proxy models, even those that are 1,000× smaller than the reference model. In the CIFAR-10 setting (cf. Figure 6 right), the LDS only greatly degrades after proxy models are more than 1,000× smaller than the reference model.

More qualitatively, we also compare the "top" and "bottom" training examples (by datamodel weight) for a given test sample in Figure 7 across proxy model sizes. Intuitively, these examples are the ones that (according to the datamodels and by linearity) most improve and most hurt, respectively, model performance if included in the training set. We find that, qualitatively, these top and bottom examples generally overlap across model scales and often have visually similar attributes. See more examples in Appendix D.

**Limitations.** We note that all the measured LDS correlations are seemingly small. The peak LDS measured in this work is roughly 0.21 for ImageNet, which indicates that we cannot exactly predict model outputs for a given training set. These LDS numbers are primarily due to (a) limitations in current datamodel

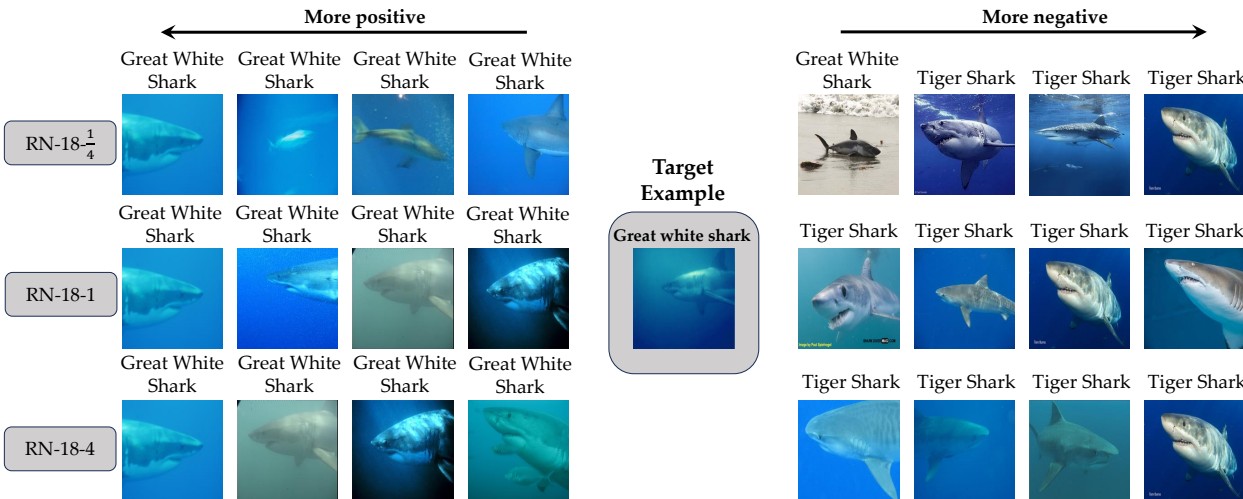

**Figure 7:** Most helpful (*left*) and most detrimental (*right*) examples for the outputs of models of different sizes are similar. The most helpful and most detrimental examples for the given target example (center) are shown according to each model size (row). We observe a large overlap between these examples. More examples in Appendix D.1.1.

estimation methods (e.g., state-of-the-art methods achieve similar LDS for CIFAR-10 (Bae et al., 2024)) and (b) inherent randomness during training[1]. The room for improvement indicates that it is possible that future, more effective datamodel estimators will behave qualitatively differently from current estimators—and that the precise trade-off between model scale and datamodel quality could change as well.

## 3.2 SELECTING TRAINING DATA WITH PROXY MODELS

In dataset selection, the goal is to choose the best possible training dataset out of a larger pool of candidate data. In this work we focus on model-aware dataset selection methods, which use the learning algorithm to select data (Xie et al., 2023a; Engstrom et al., 2024; Xia et al., 2024). Consequently, the compute cost of these methods typically grows with the cost of the learning algorithm itself[2]. As a result, model-aware dataset selection often leverage smaller proxy models for selection in place of the original (more expensive) model. In this section, we characterize the relationship between dataset selection effectiveness and proxy model size.

### 3.2.1 PRELIMINARIES

Following previous work, we formalize data selection as the problem of finding the subset of data, out of a larger pool of candidate data, that maximizes downstream trained model accuracy on a given task (Engstrom et al., 2024; Xia et al., 2024). Here, selecting training data is a supervised learning task: given (maybe only a few) samples from the test distribution, choose the data that maximizes trained model performance. In this work, we select training data with (Engstrom et al., 2024), a method that uses datamodels to select data (Ilyas et al., 2022). We refer the reader to Appendix A.4 for more details on .

A major challenge with this approach is the compute required to calculate the datamodels for language models with even as few as 1B parameters. To reduce the compute cost, Engstrom et al. (2024) computed the datamodels for a smaller proxy model and used these datamodels to select the training subset. We explore in this section the tradeoff between the scale of the proxy model to attribute and the performance of the large reference model trained on the training subset selected using the datamodels of the proxy model.

---

[1]Computing LDS requires retraining models on different subsets, and the inherent randomness involved in retraining models results in an irreducible error.

[2]In comparison, model-free dataset selection methods clean data without considering the model, instead using e.g., heuristics that capture intuitive notions of data quality (Li et al., 2024).

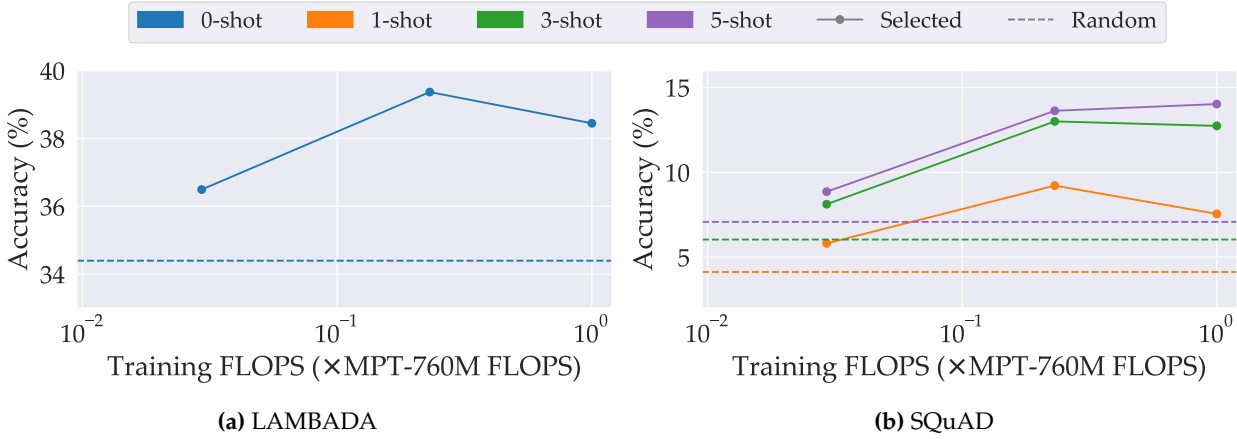

**Figure 8:** In both plots, the *x*-axis represents the amount of compute required to train a given proxy model (relative to training the large model) and the *y*-axis represents the accuracy on **(a)** LAMBADA (Paperno et al., 2016) and **(b)** SQuAD (Rajpurkar et al., 2016) of a large model trained on a subset of the MPT dataset (MosaicML, 2023a) selected using the attribution scores of a smaller model. The dashed line corresponds to the accuracy of a large model trained on a random subset of the same size as the selected dataset. Note that the training cost is only a fraction of the total attribution cost; see Appendix A.3.

### 3.2.2 Experimental Results

We study how the size of the small proxy model used for dataset selection affects model performance on two downstream tasks, SQuAD and LAMBADA.

**Setup.** We consider a language modeling (LM) setting where GPT-2 style LMs (Radford et al., 2019) are pretrained on subsets of the MPT dataset (MosaicML, 2023a)[3] and evaluated on two popular zero/few-shot classification tasks: SQuAD (Rajpurkar et al., 2016) and LAMBADA (Paperno et al., 2016).

Our large reference model is a 760M parameter LM[4], and our proxy model sizes range from 125M parameters to 760M parameters. We train all models on datasets sized according to Chinchilla-optimal token-to-parameter ratio (Hoffmann et al., 2022)[5]. We calculate the datamodels for each of our proxy models, then select a subset of the training dataset (using DSDM (Engstrom et al., 2024)) to pretrain the 760M parameter reference model. As selection baselines, we consider reference models trained on randomly-selected subsets with size dictated by the Chinchilla-optimal token-to-parameter ratio. More details are included in Appendix C.2.

**Results.** Models trained on data selected with DSDM greatly improves over those trained on randomly selected data, regardless of proxy model size (see Figure 8). We find that this improvement in downstream performance does not drop until the proxy model training scale reduces to 4x less compute than the reference model. Our results indicate that smaller proxy models mimic the behavior of reference models enough to effectively select data, while simultaneously reducing the compute cost.

## 4 Related Work

**Using smaller proxy models.** Small-scale *proxy models* are a standard building block in approaches that require understanding the role of data in large-scale models. Proxy models are used to select and clean data (Xie et al., 2023a; Engstrom et al., 2024; Chen et al., 2023; Yu et al., 2024; Li et al., 2024). At a high level, these approachs train small-scale proxy models on candidate data distributions, then analyze the resulting behavior to select the training data for the large-scale models.

---

[3]Our subset of the MPT dataset contains 160B tokens.
[4]This model is the largest we can study in our available, academic-level compute budget.
[5]We use the *llm-foundry* repository (MosaicML, 2023b) for training and evaluating our models.

**Data attribution.** Data attribution has received increased interest lately. We discuss a few of these approaches in this section. For an extensive survey of prior work, we refer the reader to (Hammoudeh & Lowd, 2022b). Some of the earliest approaches proposed the use of *influence functions* to approximate the effect of removing data points from the training dataset on a given parameter, without re-estimating the parameter (Hampel et al., 2011; Koh & Liang, 2017). Feldman & Zhang (2020); Ilyas et al. (2022) propose instead estimating empirically the effect of training data points on the model output by training several models on different subsets of the data and observing how the model output changes. Few other works have proposed different approaches to estimating these influences such as using Shapley values (Ghorbani & Zou, 2019; Jia et al., 2019; Wang et al., 2021; Shapley, 1951), gradient-based approaches (Park et al., 2023; Pruthi et al., 2020) or representational similarity (Yeh et al., 2018; Charpiat et al., 2019).

**Similarities between models trained on the same dataset.** A recent line of work argued that the data has a strong role in shaping the behavior of the trained models. Li et al. (2015) measured the extent to which multiple networks learn the same set of features, while Hermann & Lampinen (2020) studied how different models learn easy and hard features from a given dataset. Nguyen et al. (2021) on the other hand focused on how increasing the width of a network affects the learned representations. More recently, Vyas et al. (2023) investigated how increasing the width changes the properties of a model and its predictions at the example level.

**Relation between model behavior and size.** Recent work argued that the behavior of large models is predictable from smaller models under certain conditions (Yang & Hu, 2020; Yang et al., 2023). Specifically, Yang & Hu (2020) propose a parameterization of models, called $\mu P$ that guarantees the output of a model converges as its size increases. $\mu P$ has been very useful in practical setups, especially in ensuring good hyperparameters found using small models can be transferred to large models (Yang et al., 2022). Another work has argued that "emergent" abilities of large models are a mirage (Schaeffer et al., 2023) and that the reason behind the emergence can be attributed to using *hard* metrics to measure emergence (e.g., accuracy) rather than softer metrics (e.g., loss).

## 5 CONCLUSION

In this work, we argue that the the choice of training data distribution generally affects models across scale similarly, even when the difference in compute is large (175× in our experiments). This trend, however, does not always hold. In particular, given a large reference model and a *much* smaller proxy model, we identify settings where the proxy model predictions do not correlate well with the predictions of the reference model. We then study the role of proxy model size in two downstream applications: data attribution (vision setting) and dataset selection (language setting). In both settings, proxy models are (up to a certain relative scale) effective at approximating the behavior of larger models.

Taking a broader view, many important questions in machine learning reduce to understanding how changes in training setup (such as training dataset) affect the behvaior of large scale models. Small proxy models can be a powerful tool for practically and effectively answering such questions.

## ACKNOWLEDGMENT

Work supported in part by the NSF grant DMS-2134108 and Open Philanthropy.

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

APPENDICES

# A ADDITIONAL BACKGROUND

In this appendix, we present a more extensive background on datamodels (Ilyas et al., 2022) and the corresponding TRAK estimator (Park et al., 2023). We also present an extensive analysis of the compute requirement for attributing models using TRAK (Park et al., 2023). We finally present how datamodels (Ilyas et al., 2022) could be used to select optimal training sets (Engstrom et al., 2024).

**Notation.** Recall that the training set $S = \{z_1, \dots, z_n\} \subset \mathcal{Z}$ is a collection of training examples $z_i$ that could be image-label pairs or text samples. Let $L(z; \theta)$ represent the loss of a model with parameters $\theta$ on the example $z_i$. Our models are trained to minimize the empirical risk on the training set, i.e., the parameters $\theta^*(S)$ are computed as follows:

$$\theta^*(S) := \arg\min_\theta \sum_{z_i \in S} L(z_i; \theta). \tag{3}$$

The goal of *data attribution* is to trace back a model's prediction to the training data points. Formally, given an example $z$, a training dataset $S$, and a model output function $f(z; \theta)$, a data attribution function $\tau(z; S)$ is function $\tau : \mathcal{Z} \times \mathcal{Z}^n \rightarrow \mathbb{R}^n$ that maps the example $z$ and the training dataset $S$ to a real-valued score vector, called the attribution scores, where the $i^{th}$ entry corresponds to the overall importance of the training example $z_i$ on the model output $f(z; \theta^*(S))$.

## A.1 DATAMODELS

### A.1.1 INTUITION

As presented in Section 3.1.1 of the main paper, datamodels are a tool to approximate how the model output changes when trained on some subset $S'$ of the training set $S$ (Ilyas et al., 2022). Specifically, given a model with parameters $\theta^*(S')$ trained on a training subset $S'$, the goal of datamodels is to approximate how the model output $f(z; \theta^*(S'))$ on example $z$ changes for different subsets $S'$ of the training set $S$. The model output function could represent the loss of the model on the example $z$ or any other metric of interest[6].

The model output function $f(z; \theta^*(S'))$ is complex to analyze as it involves training a model on the subset $S'$ and then evaluating the resulting model on the example $z$. Instead, Ilyas et al. (2022) propose approximating this complex function $f(z; \theta^*(S'))$ using a simpler *surrogate function* $g(S')$ (Sacks et al., 1989) that doesn't involve training a new model. In practice, linear surrogate functions of the form provided a reasonable approximation of the model output (Ilyas et al., 2022; Saunshi et al., 2023). In particular, for a subset $S'$ of $S$, let $\mathbf{1}_{S'} \in \mathbb{R}^n$ be the indicator vector of $S'$ in $S$, i.e.,

$$\left(\mathbf{1}(S')\right)_j = \begin{cases} 1 & \text{if } z_j \in S' \\ 0 & \text{otherwise} \end{cases} \tag{4}$$

and let $w_{DM} \in \mathbb{R}^n$ be a *datamodel vector* (which we explain later how to compute). Ilyas et al. (2022) propose the linear surrogate function

$$g(S') := \mathbf{1}_{S'}^\top w_{DM} \tag{5}$$

to approximate the model output function $f(z; \theta^*(S'))$. The attribution scores are defined as $\tau_{DM}(z; S) = w_{DM}$.

### A.1.2 COMPUTING THE DATAMODEL VECTOR $w_{DM}$

A good datamodel vector $w_{DM}$ is one that leads to a surrogate function that approximates well the model output function $f(z; \theta^*(S'))$. When a compute is not an issue, we can search for such a vector using an optimization program that optimizes directly for our goal (good output predictability). This can be achieved as follows:

**Step 1.** Sample at random $M$ training subsets $\{S_i : S_i \subset S\}_{i=1}^M$ and collect their indicator vectors $\{\mathbf{1}_{S_i}\}_{i=1}^M$.

**Step 2.** Train a model on each subset $S_i$ and collect model parameters $\{\theta^*(S_i)\}_{i=1}^M$.

**Step 3.** Compute the output of each model for example $z$, i.e., $\{f(z; \theta^*(S_i))\}_{i=1}^M$.

---

[6]We have presented two different examples of model output functions in the main paper.

**Step 4.** Compute the datamodel vector $w_{DM}$ by regression on the dataset $\left\{ \left( \mathbf{1}_{S_i}, f(z; \theta^*(S_i)) \right) \right\}_{i=1}^{M}$.

The regression over the dataset $\left\{ \left( \mathbf{1}_{S_i}, f(z; \theta^*(S_i)) \right) \right\}_{i=1}^{M}$ is usually performed using LASSO (Ilyas et al., 2022; Tibshirani, 1994), i.e.,

$$w_{DM} = \arg\min_{w} \frac{1}{M} \sum_{i=1}^{M} \left( \mathbf{1}_{S_i}^{\top} w_{DM} - f(z; \theta^*(S_i)) \right)^2 + \beta \cdot \|w\|_1. \tag{6}$$

This procedure produces a datamodel vector $w_{DM}$ that could be used in the context of the surrogate function $g$ to estimate the output $f(z; \theta^*(S'))$ of a model trained on the subset $S'$, without training the model on $S'$. In the context of data attribution, the datamodels attribution scores correspond to the datamodel vector, i.e., $\tau_{DM}(z, S) = w_{DM}$. We present the full procedure in Algorithm 1.

---

**Algorithm 1** Computing the datamodel vector $w_{DM}$

---

**Require:** Target example $z$, dataset $S = \{z_i\}_{i=1}^{n}$ with $n$ samples, subset ratio $\alpha$, number of models $M$, regularization parameter $\beta$
1: Sample $M$ random subsets $S_1, S_2, \ldots, S_M \subset S$ of size $\lfloor \alpha \cdot n \rfloor$
2: **for** $i \in 1$ to $M$ **do**
3:     Record indicator vector $\mathbf{1}_{S_i}$
4:     Train model on $S_i$ and collect parameters $\theta^*(S_i)$
5:     Record the model output function $f(z; \theta^*(S_i))$
6: **end for**
7: Compute datamodel vector $w_{DM}$ as:

$$w_{DM} = \arg\min_{w} \frac{1}{M} \sum_{i=1}^{M} \left( \mathbf{1}_{S_i}^{\top} w_{DM} - f(z; \theta^*(S_i)) \right)^2 + \beta \cdot \|w\|_1.$$

8: **return** $w_{DM}$

---

## A.2 Approximating Datamodels with TRAK

In the following section, we present how TRAK (Park et al., 2023) provides an efficient estimate of datamodels (Ilyas et al., 2022). For a more extensive analysis, please refer to the TRAK paper (Park et al., 2023).

### A.2.1 Intuition

Computing the attribution scores using datamodels is an expensive process (Ilyas et al., 2022) as it involves training a large number of models $M$ on subsets of the training dataset. This approach is not feasible beyond simple toy settings. To reduce the computational requirement, Park et al. (2023) propose approximating datamodels by first casting the problem into a logistic regression setup, and then computing the attribution scores efficiently in this new regime. At a high level, casting the original problem into a regression setup can be done by representing the model at hand using a kernel machine (Jacot et al., 2018). Once the problem is cast into this simple form, prior work has developed a closed-form solution for data attribution in a logistic regressing setup (Pregibon, 1981). Below, we first present the solution for the logistic regression setup and then present how to cast classification with neural networks into this linear setup.

### A.2.2 Approximating Datamodels in a Logistic Regression Setup

We borrow notation from (Park et al., 2023) and refer the readers to the paper for a more extensive analysis. Consider a logistic regression setup where we have a dataset $S = \{z_1, \ldots, z_n\}$ where each example $z_i = (x_i, b_i, y_i)$ is triple of an input $x_i \in \mathbb{R}^d$, a bias term $b_i \in \mathbb{R}$ and a label $y_i \in \{-1, 1\}$.

In this setup, we can formulate the logistic regression problem:

$$\theta^*(S) := \arg\min_{\theta} \sum_{i} \log \left[ 1 + \exp(-y_i \cdot (x_i^{\top} \theta + b_i)) \right]. \tag{7}$$

In this simple setup, we define our model output function as the logit function: $f(z;\theta) := x^\top\theta + b$, where $z = (x, b, y)$.

The problem of data attribution in this simple setup is well-studied in literature, and prior work has developed a closed-form solution for it (Pregibon, 1981). In particular, the contribution of a training example $z_i$ to the model output function $f(z;\theta)$ can be measured using the *leave-one-out* influence (LOO) (Pregibon, 1981), described below:

$$\tau_{LOO}(z, S) := \frac{x^\top(X^\top RX)^{-1}x_i}{1 - x_i^\top(X^\top RX)^{-1}x_i \cdot p_i^* \cdot (1 - p_i^*)} \cdot (1 - p_i^*) \approx f(z;\theta^*(S)) - f(z, \theta^*(S\backslash\{z_i\})), \quad (8)$$

where $X \in \mathbb{R}^{n\times d}$ is the matrix of stacked inputs $x_i$, and $p_i^* = [1 + \exp(-y_i \cdot f(z_i;\theta^*))]^{-1}$ is predicted probability of the correct class, $R \in \mathbb{R}^{n\times n}$ is a diagonal matrix where $R_{ii} = p_i^* \cdot (1 - p_i^*)$, and $S\backslash\{z_i\}$ is the training set without example $z_i$. This influence score approximates the effect of removing training example $z_i$ from the training dataset.

In practice, computing the attribution scores in a logistic regression setup using this closed-form solution is efficient and fast. Many interesting problems in ML, however, are highly non-linear. In the next section, we show how we can cast a non-linear problem using neural networks into linear regression problems.

### A.2.3 CASTING NON-LINEAR PROBLEMS INTO LOGISTIC REGRESSION

In this section, we first start by considering a non-linear binary regression setup. We then present how to generalize the approach to multi-class classification and language modeling.

Given a non-linear binary regression setup, we can express the parameters of the model trained on the dataset as:

$$\theta^*(S) := \arg\min_\theta \sum_i \log\left[1 + \exp(-y_i \cdot f(z_i;\theta))\right]. \quad (9)$$

The main challenge in this setup is the non-linearity in the model output function $f(z;\theta)$. Park et al. (2023) propose to solve this problem by casting the problem at hand into a linear problem. Specifically, given a neural network with model output function $f(z;\theta)$, the authors approximate the model output function around the parameters $\theta^*$ of the optimal model using a Taylor's approximation:

$$\hat{f}(z;\theta) := f(z;\theta^*) + \nabla_\theta f(z;\theta^*)^\top(\theta - \theta^*). \quad (10)$$

This step corresponds in the literature to replacing the binary classifier with its eNTK approximation (Jacobsen et al., 2018; Atanasov et al., 2022; Wei et al., 2022a). Given this linearization, we adapt Equation (9) and write instead:

$$\theta^*(S) := \arg\min_\theta \sum_i \log\left[1 + \exp(-y_i \cdot f(z_i;\theta))\right] \quad (11)$$

$$:= \arg\min_\theta \sum_i \log\left[1 + \exp\left(-y_i \cdot \left(f(z_i;\theta^*) + \nabla_\theta f(z_i;\theta^*)^\top(\theta - \theta^*)\right)\right)\right] \quad (12)$$

$$:= \arg\min_\theta \sum_i \log\left[1 + \exp\left(-y_i \cdot \left(\nabla_\theta f(z_i;\theta^*)^\top\theta + f(z_i;\theta^*) - \nabla_\theta f(z_i;\theta^*)^\top\theta^*\right)\right)\right] \quad (13)$$

$$:= \arg\min_\theta \sum_i \log\left[1 + \exp\left(-y_i \cdot \left(g_i^\top\theta + b_i\right)\right)\right], \quad (14)$$

where the vector $g_i := \nabla_\theta f(z_i;\theta^*)$ corresponds to the model gradients and we define the bias term $b_i := f(z_i;\theta^*) - \nabla_\theta f(z_i;\theta^*)^\top\theta^*$.

The form we observe in Equation (14) is reminiscent of Equation (7). In fact, given our examples $z_i = (g_i, b_i, y_i)$, we can apply in closed-form the solution from Equation (8) to compute the attribution scores. However, one big issue in practice is the large dimensionality of the vector $g_i$, which corresponds to the number of model parameters. This value could be in the billions for the largest available models and as such estimating the attribution scores using Equation (8) is intractable.

### A.2.4 REDUCING THE DIMENSIONALITY AND ESTIMATING DATAMODELS

Given the intractability of the problem, Park et al. (2023) propose reducing the dimensionality of the gradient vectors $g_i$ using random projections (Johnson & Lindenstrauss, 1984). While many techniques exist for reducing the dimensionality of a vector, the authors choose random projections since they preserve some desired properties in the logistic regression problem. We refer the readers to (Park et al., 2023) and (Malladi et al., 2022) for more details on this choice.

Given a vector $g \in \mathbb{R}^p$ and a random matrix $\mathbf{P} \in \mathbb{R}^{k \times p}$, where $k \ll p$, we define the feature map $\phi$ : $\mathbb{R}^p \to \mathbb{R}^k$ as $\phi(g) = \mathbf{P}^\top g$. With this feature map, we project all gradients $g_i$ to obtain feature vectors $\phi_i = \phi(g_i) = \mathbf{P}^\top g_i$, and stack them into the matrix $\Phi := [\phi_1, \ldots, \phi_n] \in \mathbb{R}^{n \times k}$. Notice how this matrix is much smaller than the original matrix $X = [g_1, \ldots, g_n] \in \mathbb{R}^{n \times p}$.

Using the matrix $\Phi$ of stacked gradients, we can compute the attribution scores as:

$$\tau_{TRAK}(z, S) = \phi(z)^\top (\Phi^\top \Phi)^{-1} \Phi^\top \mathbf{Q}, \tag{15}$$

where $\phi(z) = \mathbf{P}^\top \nabla_\theta f(z; \theta^*)$ corresponds to the projected gradient of the target example $z$, and the matrix $\mathbf{Q} := \text{diag}(\{1 - p_i^*\}_i)$ is a diagonal matrix with the probabilities of the correct class $p_i^* = [1 + \exp(-y_i \cdot f(z_i; \theta^*))]^{-1}$. Park et al. (2023) find that dropping the matrix $R$ and the denominator do not affect the predictiveness of the attribution scores. For more details, we refer the readers to the paper (Park et al., 2023).

### A.2.5 IMPROVING THE DATAMODELS ESTIMATION USING ADDITIONAL MODELS

One main challenge with the previous approach is the stochastic nature of training models. In particular, changing the random seed and training the same model on the same dataset can lead to widely different results across multiple runs (Nguyen et al., 2021; D'Amour et al., 2020). To solve this problem, Park et al. (2023) propose training $M$ models and then averaging across multiple runs as follows:

$$\tau_{TRAK}(z, S) = \left( \frac{1}{M} \sum_{m=1}^{M} \phi_m(z)^\top (\Phi_m^\top \Phi_m)^{-1} \Phi_m^\top \right) \cdot \left( \frac{1}{M} \sum_{m=1}^{M} \mathbf{Q}_m \right), \tag{16}$$

where the feature map and vectors are different for each of the $M$ runs. Notice that the authors average across the feature maps rather than over attribution scores for numerical stability reasons (Park et al., 2023).

In this work, we propose a further modification where we drop the term corresponding to the matrix $\mathbf{Q}_m$ from our estimator. Specifically, we compute the attribution scores as:

$$\tau_{TRAK}(z, S) = \frac{1}{M} \sum_{m=1}^{M} \phi_m(z)^\top (\Phi_m^\top \Phi_m)^{-1} \Phi_m^\top \tag{17}$$

$$= \frac{1}{M} \sum_{m=1}^{M} \tau_{TRAK}^{(m)}(z, S). \tag{18}$$

We notice that dropping the last term does not affect negatively the predictiveness of the attribution scores, and can in many cases in practice improve it. In particular, for many models, the pre-softmax logit can be very large and saturates the softmax when computing probabilities, which in turn leads to multiple 0 entries in the matrix $\mathbf{Q}_m$ and consequently the attribution scores. This behavior reduces drastically the counterfactual predictability, measured using the LDS.

### A.2.6 GENERALIZING TO MULTI-CLASS CLASSIFICATION

In the previous sections, we presented how to cast general non-linear binary classification problems into a linear regression setup in order to estimate the attribution scores efficiently. In this section, we show how Park et al. (2023) extended the previous approach to support general multi-class classification setups.

Given a multi-class classification problem over $c$ classes, let $p(z; \theta)$ be the probability assigned by the model to the *correct* class. Park et al. (2023) define the model output function in this setup to be:

$$f(z; \theta) = \frac{p(z; \theta)}{1 - p(z; \theta)}. \tag{19}$$

This model output function essentially measures whether the correct class is more likely than any other class[7]. One nice property of this model output function is that it allows to write the loss function $L(z; \theta)$ as follows:

$$L(z; \theta) = -\log(p(z; \theta)) \tag{20}$$
$$= \log\left[1 + \exp\left(-f(z; \theta)\right)\right], \tag{21}$$

which is reminiscent of Equation (9) (with $y_i = 1$). As such, we can make the same approximations made in the binary case setup and apply the same results and derivations to compute the attribution scores. We present the full procedure in Algorithm 2.

---

**Algorithm 2** Approximating the datamodel vector using TRAK for multi-class classification

---

**Require:** Target example $z$, dataset $S = \{z_i\}_{i=1}^n$ with $n$ samples, number of models $M$, correct-class likelihood $p(z; \theta)$, projection dimension $k \in \mathbb{N}$

1: Define model output function: $f(z; \theta) := \frac{p(z; \theta)}{1 - p(z; \theta)}$
2: **for** $m \in 1$ to $M$ **do**
3:     Train model with parameters $\theta_m^*(S)$ on dataset $S$
4:     Sample projection matrix $\mathbf{P}_m \sim \mathcal{N}(0, 1)^{n \times k}$
5:     **for** $i \in 1$ to $n$ **do**
6:         Compute gradient and project: $\phi_i = \mathbf{P}_m^\top \nabla_\theta f(z_i; \theta_m^*(S))$
7:     **end for**
8:     Stack projected gradients: $\Phi_m = [\phi_1, \ldots, \phi_n]^\top$
9: **end for**
10: Compute the attribution scores using:

$$\tau_{TRAK}(z, S) = \frac{1}{M} \sum_{m=1}^M \phi_m(z)^\top (\Phi_m^\top \Phi_m)^{-1} \Phi_m^\top$$

11: **return** $\tau_{TRAK}(z, S)$

---

### A.2.7 ADAPTING THE TRAK ESTIMATOR TO LANGUAGE MODELS

So far, we have presented how TRAK (Park et al., 2023) could be applied for classification setups. We now present how TRAK could be extended to support language models, as presented in (Engstrom et al., 2024).

Recall that for multi-class classification, Park et al. (2023) define the model output function to be:

$$f(z; \theta) = \frac{p(z; \theta)}{1 - p(z; \theta)}, \tag{22}$$

where $p(z; \theta)$ is the probability of the correct class. This setup can be naturally extended to language models trained based on next-token prediction (Sutskever et al., 2014; Vaswani et al., 2017) where the goal is to iteratively predict out of many tokens the correct token to continue the sentence. Specifically, given a sequence $z = \{z_1, \ldots, z_T\}$ of context length $T$, let $p(z_j \mid z_{<j}; \theta)$ be the probability of predicting the correct token at position $j$ of the sequence, given the previously predicted tokens $z_1, \ldots, z_{j-1}$. This prediction is applied $T - 1$ times, with each occurrence being its own classification problem. We can then define the it language-modeling model output function as the *average* model output function across all classification tasks (Engstrom et al., 2024):

$$f(z; \theta) = \frac{1}{T} \sum_{j=2}^T \frac{p(z_j \mid z_{<j}; \theta)}{1 - p(z_j \mid z_{<j}; \theta)}. \tag{23}$$

With this new definition, we can apply the TRAK framework (Park et al., 2023) as outlined in Algorithm 2.

---

[7]This is more tractable than defining $c^2$ classification problems between all pairs of classes.

### A.3    ESTIMATING COMPUTE REQUIREMENT

In this section, we give an overview of the overall compute requirement. Our analysis focuses mostly on the language setup, where we have observed that compute is a bigger bottleneck. A similar analysis could be done for our vision setup.

#### A.3.1    COST TO TRAIN A SINGLE MODEL

We assume the models being trained are transformers (Vaswani et al., 2017) and leverage the compute approximations presented in (Kaplan et al., 2020)[8]. Specifically, given a transformer model with $p$ parameters and a dataset composed of $D$ tokens ($n_{train}$ examples[9] with $T$ tokens each), the total cost (measured in FLOPS) for training the transformer on the dataset can be approximated as

$$C^{train} = C^{forward} + C^{backward} \tag{24}$$
$$= 2pD + 4pD \tag{25}$$
$$= 6p \cdot T \cdot n_{train}. \tag{26}$$

#### A.3.2    COST TO ATTRIBUTE A SINGLE MODEL

As outlined in the previous sections, the attribution scores (using a single model) on a single target example can be computed using:

$$\phi(z)^\top (\Phi^\top \Phi)^{-1} \Phi^\top, \tag{27}$$

where $\phi(z) \in \mathbb{R}^k$ is the projected gradient of the target example $z$ and $\Phi \in \mathbb{R}^{n \times k}$ is the stacked matrix of projected inputs, $n$ is the total number of training examples and $k$ is the projection dimension. We assume the cost for multiplying matrices $A \in \mathbb{R}^{a \times b}$ and $B \in \mathbb{R}^{b \times c}$ to be $a \cdot c \cdot (2b - 1)$ FLOPS.

We can break down our costs as follows:

1. The cost to compute the gradients for the training set is $6pD = 6p \cdot T \cdot n$.

2. The cost to compute the gradients for the target example is $6p$. When dealing with a target dataset with $n_{test}$ examples, this cost is $6p \cdot T \cdot n_{test}$.

3. The cost to randomly project the gradients of the training examples is $n \cdot k \cdot (2p - 1)$.

4. The cost to randomly project the gradients of the test examples is $n_{test} \cdot k \cdot (2p - 1)$.

5. The product $\Phi^\top \Phi$ requires $k^2 \cdot (2n - 1)$ FLOPS.

6. The inverse operation $(\Phi^\top \Phi)^{-1}$ costs around $k^3$ FLOPS.

7. The product $(\Phi^\top \Phi)^{-1} \Phi^\top$ costs $n \cdot k \cdot (2k - 1)$ FLOPS.

8. The final product $\phi(z)^\top (\Phi^\top \Phi)^{-1} \Phi^\top$ costs $n \cdot (2k - 1)$ FLOPS for a single target example $z$, and $n_{test} \cdot n \cdot (2k - 1)$ for attributing over $n_{test}$ target examples.

The total attribution cost is then the sum of the above terms:

$$C^{attrib} = (6pT + 4k^2 + 2k \cdot p - 2k + 2k \cdot n_{test} - n_{test}) \cdot n \tag{28}$$
$$+ 6p \cdot T \cdot n_{test} + k \cdot (2p - 1) \cdot n_{test} - k^2 + k^3 \tag{29}$$
$$\approx (6p \cdot T + 4k^2 + 2k \cdot p) \cdot n + 2p \cdot (3T + k) \cdot n_{test} + k^3. \tag{30}$$

---

[8]Better approximations exist (Hoffmann et al., 2022), but they do not lead to substantially different approximations.
[9]Given a very large dataset with a total of $n$ examples, compute optimal models can usually be trained using a much smaller number of training examples $n_{train}$(Hoffmann et al., 2022).

### A.3.3 OVERALL COST

Using our previous estimates, we can estimate the overall cost as:

$$C^{total} = C^{train} + C^{attrib} \tag{31}$$

$$= 6p \cdot T \cdot n_{train} + (6p \cdot T + 4k^2 + 2k \cdot p) \cdot n + 2p \cdot (3T + k) \cdot n_{test} + k^3 \tag{32}$$

$$= \left(6p \cdot T \cdot \left(1 + \frac{n_{train}}{n}\right) + 4k^2 + 2k \cdot p\right) \cdot n + 2p \cdot (3T + k) \cdot n_{test} + k^3. \tag{33}$$

Asymptotically, we find that the ratio of the training cost to the overall cost is:

$$\frac{C^{train}}{C^{total}} = \frac{6p \cdot n_{train} \cdot T}{\left(6p \cdot T \cdot \left(1 + \frac{n_{train}}{n}\right) + 4k^2 + 2k \cdot p\right) \cdot n + 2p \cdot (3T + k) \cdot n_{test} + k^3} \tag{34}$$

$$\rightarrow \frac{3 \cdot T}{6 \cdot T + k} \tag{35}$$

$$\approx 22.22\% \quad \text{(for our setup)}, \tag{36}$$

assuming very large compute-optimal models where $n_{train} = n$ (Hoffmann et al., 2022). We present an example of our compute estimates in Table 9.

Note that we use $M$ models to improve our attribution scores computed using TRAK (Park et al., 2023). This increases all our cost estimates by a factor of $M$.

**Table 9:** Compute requirement for attributing our different MPT models (MosaicML, 2023b).

| Parameter | MPT-125M | MPT-350M | MPT-760M | MPT-8B[10] |
|---|---|---|---|---|
| $p$ ($\times 10^6$ params) | 125 | 350 | 760 | 8000 |
| $n_{train}$ ($\times 10^6$ examples) | 1.33 | 3.68 | 7.47 | 80 |
| $n$ ($\times 10^6$ examples) | 80 | 80 | 80 | 80 |
| $n_{test}$ (examples) | 1,000 | 1,000 | 1,000 | 1,000 |
| $T$ (tokens) | 2,048 | 2,048 | 2,048 | 2,048 |
| $k$ (proj dim) | 15,360 | 15,360 | 15,360 | 15,360 |
| $C^{train}$ (FLOPS) | $2.04 \times 10^{18}$ | $1.58 \times 10^{19}$ | $6.97 \times 10^{19}$ | $7.86 \times 10^{21}$ |
| $C^{attrib}$ (FLOPS) | $4.30 \times 10^{20}$ | $1.20 \times 10^{21}$ | $2.62 \times 10^{21}$ | $2.75 \times 10^{22}$ |
| $C^{overall}$ (FLOPS) | $4.32 \times 10^{20}$ | $1.22 \times 10^{21}$ | $2.68 \times 10^{21}$ | $3.53 \times 10^{22}$ |
| $\frac{C^{train}}{C^{overall}}$ (%) | 0.47 | 1.29 | 2.60 | 22.22 |
| $\frac{p}{p\,(\text{MPT-125M})}$ | 1.00 | 2.80 | 6.08 | 64.00 |
| $\frac{C^{overall}}{C^{overall}\,(\text{MPT-125M})}$ | 1.00 | 2.82 | 6.21 | 81.88 |

### A.3.4 PRACTICAL CONSIDERATIONS

In the previous section, we focused solely on the asymptotic behavior. Even in that regime, the boost from using smaller models is already super-linear. In real life, other practical considerations would emerge. For example, models of different sizes might require different amounts of GPU memory, which in turn affects the number of parallel operations within the TRAK framework (Park et al., 2023). Other considerations include the network bandwidth, especially since we are dealing with massive datasets of several terabytes. All these factors affect our compute estimates and the speedup. The results in Table 9 merely reflect a lower bound on the speedups in realistic setups.

## A.4 DATASET SELECTION WITH DATAMODELS (DSDM)

In this section, we present additional background on the downstream application of data attribution that we consider: dataset selection (Brown et al., 2020; Xie et al., 2023b; Engstrom et al., 2024). We focus on the setup adopted in (Engstrom et al., 2024). For more details, we refer the reader to the paper (Engstrom et al., 2024).

### A.4.1 PROBLEM SETUP

Dataset selection refers to the task of selecting from a large pool of data a training set that leads to the "best" performance on a given target task. Engstrom et al. (2024) cast the dataset selection task into an optimization problem where the objective function is the model loss on a target task and the decision variable is the dataset selected from a large pool of data.

More precisely, given a large pool of data $\mathcal{Z}$, a target distribution $\mathcal{D}_{\text{targ}}$ (e.g., a language modeling task) and a target dataset size $n$, we can formulate the dataset selection task as:

$$S^* := \arg\min_{\substack{S \subset \mathcal{Z} \\ |S|=n}} \mathcal{L}_{\mathcal{D}_{\text{targ}}}(S) \tag{37}$$

$$:= \arg\min_{\substack{S \subset \mathcal{Z} \\ |S|=n}} \mathbb{E}_{z \sim \mathcal{D}_{\text{targ}}}\left[L(z; \theta^*(S))\right] \tag{38}$$

where $\theta^*(S)$ are the parameters of the model trained on $S$, $L(z; \theta^*(S))$ is the loss achieved by the model on target example $z \sim \mathcal{D}_{\text{targ}}$ and $\mathcal{L}_{\mathcal{D}_{\text{targ}}}(S)$ is the expected loss of the models trained on $S$ on samples from the target distribution $\mathcal{D}_{\text{targ}}$.

### A.4.2 APPROXIMATING SOLUTION WITH DATAMODELS

The optimization problem in Equation (38) is generally hard to solve as it involves a combinatorial search over $\binom{|\mathcal{Z}|}{n}$ possible solutions. Furthermore, evaluating each candidate solution $S$ requires training a new model on the chosen training set $S$ then measuring the model's loss on the target task.

To circumvent this problem, Engstrom et al. (2024) propose using datamodels (Ilyas et al., 2022) to approximate the loss of the model trained on the candidate solution $S$. An additional advantage of this approach is the linear relationship between the indicator vector of the set $S$ and the target loss (see Equation (5)), which makes the optimization problem easier.

Recall that for a given example $z$, datamodels approximate the complex model output function $f(z; \theta^*(S))$ using a linear surrogate function $g(S) = \mathbf{1}_S^\top w_z$, where $w_z \in \mathbb{R}^{|\mathcal{Z}|}$ is the datamodel vector corresponding to target example $z$[11]. Using the linear surrogate function, we can approximate for a candidate set $S$ the model's expected loss as:

$$\mathbb{E}_{z \sim \mathcal{D}_{\text{targ}}}\left[L(z; \theta^*(S))\right] \approx \mathbb{E}_{z \sim \mathcal{D}_{\text{targ}}}\left[\mathbf{1}_S^\top w_z\right] \tag{39}$$

$$= \mathbf{1}_S^\top \mathbb{E}_{z \sim \mathcal{D}_{\text{targ}}}\left[w_z\right] \tag{40}$$

$$\approx \mathbf{1}_S^\top \left(\frac{1}{m}\sum_{i=1}^{m} w_{z_i}\right) \tag{41}$$

---

[11]We refer to the datamodel vector $w_{DM}$ as $w$ for ease of notation.

where we assume we have access to $m$ samples from the target distribution $\mathcal{D}_{\text{targ}}$. With this approximation, we rewrite the optimization program from Equation (38) as:

$$S^* := \arg\min_{\substack{S \subset \mathcal{Z} \\ |S|=n}} \mathbb{E}_{z \sim \mathcal{D}_{\text{targ}}} \left[ L(z; \theta^*(S)) \right] \tag{42}$$

$$\approx \arg\min_{\substack{S \subset \mathcal{Z} \\ |S|=n}} \mathbf{1}_S^\top \left( \frac{1}{m} \sum_{i=1}^m w_{z_i} \right) \tag{43}$$

$$= \arg\text{bot-}n \left( \frac{1}{m} \sum_{i=1}^m w_{z_i} \right) \tag{44}$$

which corresponds to choosing the indices corresponding to the bottom $n$ values of the vector $\left( \frac{1}{m} \sum_{i=1}^m w_{z_i} \right)$.

With this new formulation, the task of dataset selection reduces to estimating the datamodels vectors for a given downstream task and then finding the indices corresponding to the bottom $n$ values of the average datamodels vector.

In practice, computing datamodels (Ilyas et al., 2022) is expensive, so Engstrom et al. (2024) approximate them using the TRAK framework (Park et al., 2023). We present an overview of the procedure in Algorithm 3.

---

**Algorithm 3** Dataset selection using datamodels (DSDM)

---

**Require:** Large pool of data $\mathcal{Z}$, selected dataset size $n$, $m$ target examples $\{z_1, \ldots, z_m\}$ from distribution $\mathcal{D}_{\text{targ}}$
  1: Estimate datamodels vectors $\{w_{z_i}\}_{i=1}^m$ from $\mathcal{Z}$ using TRAK
  2: Compute average datamodel vector $w_{\text{targ}} = \left( \frac{1}{m} \sum_{i=1}^m w_{z_i} \right)$
  3: Collect indices $\mathcal{I} = \arg\text{bot-}n \left( w_{\text{targ}} \right)$
  4: **return** optimal set $S^*$ of training examples from pool $\mathcal{Z}$ at indices $\mathcal{I}$

---

# B   SIMILARITY BETWEEN SMALL AND LARGE MODELS

In the main paper, we demonstrated that when models of different sizes are trained on the same data distribution, their losses are surprisingly linear (see Figure 1). In this section, we present additional details on the experimental setup of our result from Figure 1, and then present more results in the vision setting.

## B.1   LANGUAGE SETTING

### B.1.1   EXPERIMENTAL SETUP

**Models.**   In this setting, we consider two models based on the MPT architecture (MosaicML, 2023b): a small model with 80M parameters and a larger one with 760M parameters. The small model is trained on 1.67B tokens while the large model is trained on 15.3B tokens[12]. This makes the small model require 85x less compute than the larger model. Both models have a context length of 1,024. More architectural details in Table 10 below.

**Table 10:** The architecture of our small and large MPT models (MosaicML, 2023b) used for Figure 1.

|                      | Model Dim | Heads | Layers | Parameters  | Train Tokens (B) |
|----------------------|-----------|-------|--------|-------------|------------------|
| **MPT-80M** (*small*)  | 640       | 10    | 10     | 82,127,360  | 1.67             |
| **MPT-760M** (*large*) | 1,536     | 12    | 24     | 760,470,528 | 15.3             |

**Data distributions.**   We train several copies of the small and large models, each on a different data distribution. Some of our distributions are *natural* while the rest are induced by algorithms.

- *Natural distributions*:
  - **MPT dataset** (MosaicML, 2023a): The MPT dataset is a collection of examples from several online sources such as CommonCrawl, RedPajama, etc.[13] We train our models on random subsets from the MPT dataset.
  - **RedPajama-ArXiV** (Computer, 2023): The data consists of ArXiV articles and is extracted from the MPT subset.
  - **RedPajama-Books** (Computer, 2023): The data consists of subsets of books and is extracted from the MPT subset.
  - **RedPajama-Wiki** (Computer, 2023): The data consists of Wikipedia articles and is extracted from the MPT subset.
  - **Semantic Scholar** (Lo et al., 2020): The data consists of Semantic Scholar articles and is extracted from the MPT subset.
  - **Stack-Markdown** (Kocetkov et al., 2022): The data consists of Markdown code from the Stack dataset and is extracted from the MPT subset.
- *Algorithm-induced distributions*[14]:
  - **DsDm** (Engstrom et al., 2024): DSDM is a method for selecting pretraining examples that improve the downstream performance. We reuse the outcomes of this method when applied to the C4 dataset (Raffel et al., 2020) as presented in (Engstrom et al., 2024).
  - **Bot-DsDm** (Engstrom et al., 2024): This method is simply the reverse of DSDM. Specifically, we choose the pretraining examples that hurt performance the most. While this distribution is not particularly useful practically, it is helpful insofar as it reflects how language models behave at the other end of the spectrum.
  - **DSIR** (Xie et al., 2023b): DSIR is a method to choose pretraining examples that improve performance through importance resampling. We reuse the outcomes of this method when applied to the C4 dataset (Raffel et al., 2020) as presented in (Engstrom et al., 2024).

---

[12]The number of tokens was chosen to optimize for the compute level, as described in (Hoffmann et al., 2022).
[13]An extensive list of sources can be found at (MosaicML, 2023a).
[14]The data can be found at https://github.com/MadryLab/DsDm.

- **Classifier** (Brown et al., 2020): Classifier is a method to choose pretraining examples that improve performance by using a classifier that predicts whether the pretraining examples are similar to the downstream examples or not. We reuse the outcomes of this method when applied to the C4 dataset (Raffel et al., 2020) as presented in (Engstrom et al., 2024).

**Downstream datasets.** After training our models on each of the data distributions highlighted above, we measure their losses on several datasets. The goal is to reflect the linearity over multiple data distributions.

- **C4** (Raffel et al., 2020): This dataset consists of web-extracted text from Common Crawl during April 2019.
- **The Pile** (Gao et al., 2020): This dataset consists of text extracted from multiple sources, including Common Crawl, Books, etc. More details can be found in the paper.
- **SQuAD** (Rajpurkar et al., 2016): Stanford Question Answering Dataset (Rajpurkar et al., 2016) is a reading comprehension dataset composed of excerpts from Wikipedia articles. The task in this dataset is answering questions given some context.
- **LAMBADA** (Paperno et al., 2016): LAnguage Modeling Broadened to Account for Discourse Aspects (Paperno et al., 2016) is a dataset that measures broad context understanding through the means of word prediction. Paperno et al. (2016) collected text narratives where human annotators are able to predict the last word in a sentence when they have seen the whole passage but not when they only see the last sentence before the text completion. We use the version of the dataset cleaned by EleutherAI[15].
- **HellaSwag** (Zellers et al., 2019): HellaSwag is multiple-choice dataset extracted from the SWAG dataset (Zellers et al., 2018). The dataset is extracted using Adversarial Filtering (AF) and is challenging to language models while being almost trivial for humans.

For the downstream datasets SQuAD, LAMBADA and HellaSwag, we measure the models' losses only over their predictions, while for the pretraining datasets C4 and The Pile, we measure their losses over the whole sequence.

### B.1.2 CORRELATION AT THE EXAMPLE LEVEL

In Figure 1, we show that the losses achieved by the small and large models on a target distribution are linear. We extend this result and show that for some sequences, in the downstream tasks, the losses achieved at the example level are also linear (see Figure 4).

We then plot the coefficient of determination ($R^2$) between the losses achieved by the small and large models on target examples in each downstream task (see Figure 5). We can see that a significant proportion of the target examples have a positive $R^2$.

### B.1.3 CORRELATION FOR LARGER COMPUTE GAP

We now investigate how our results change when we increase the compute gap. To that end, we consider a smaller model consisting of 37 million parameters and trained on 840 million tokens. We consider the same large model of 760 million parameters. The difference in compute in this case is 370x. For more architectural details, check Table 11.

**Table 11:** The architecture of our small and large MPT models (MosaicML, 2023b) used for Figure 1.

|                      | Model Dim | Heads | Layers | Parameters  | Train Tokens (B) |
|----------------------|-----------|-------|--------|-------------|------------------|
| **MPT-37M** (*small*)  | 384       | 6     | 10     | 37,479,936  | 0.84             |
| **MPT-760M** (*large*) | 1,536     | 12    | 24     | 760,470,528 | 15.9             |

When the compute gap is larger, we still observe a very strong correlation between small and large models, albeit slightly weaker on some downstream tasks, e.g., SQuAD (Rajpurkar et al., 2016) (see Figure 12).

---

[15]The dataset can be found on: `https://huggingface.co/datasets/EleutherAI/lambada_openai/viewer/en`.

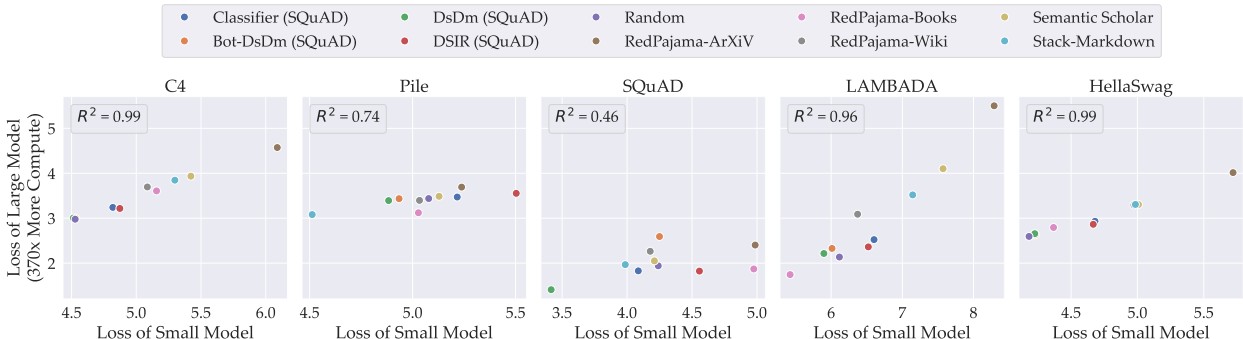

**Figure 12:** Small models are reliable proxies of large models. In all plots, the *x*-axis represents the loss achieved by a small MPT model of 37 million parameters trained on 0.84B tokens and the *y*-axis represents the loss achieved by a larger MPT model of 760 million parameters trained on 15.9B tokens. Each plot corresponds to a different target distribution, and within each plot, each point corresponds to a different training distribution.

We next investigate how this correlation changes at the example level. Similar to the earlier setting, we observe a strong (still weaker) correlation (see Figure 13 and Figure 4). These results indicate that small models can still be reliable proxies of large models, even when the difference in compute is different by orders of magnitude.

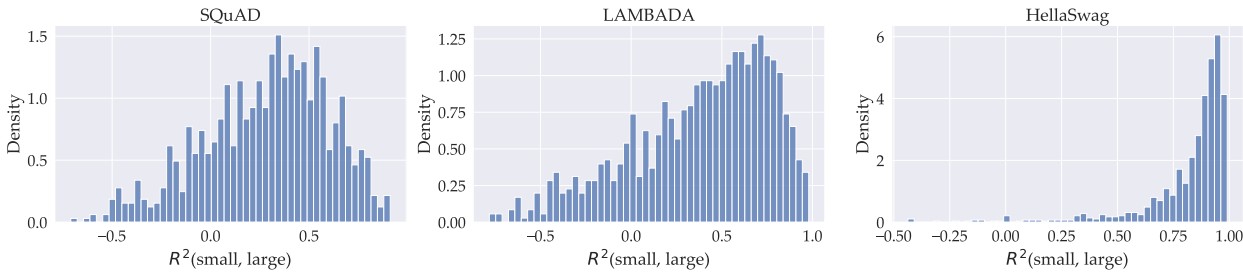

**Figure 13:** Small models are reliable proxies of large models. We plot the coefficient of determination ($R^2$) between the losses of the small and large models for all examples.

### B.1.4  How Correlation Changes with Compute

We observe a large correlation between the losses achieved by our small and large models over multiple tasks. To test the extent of this correlation, we train several other models of different sizes (125M, 220M and 350M) with different compute budgets, measure the coefficient of determination between their losses and the loss of the large model (760M) and then plot how this correlation (averaged over multiple tasks) changes as a function of the training compute budget. We see in Figure 2 that the correlation increases as the training compute increases.

### B.2  Vision Setting

We show in this section that our results still hold across the vision setting.

### B.2.1  Experimental Setup

**Models.** We consider variants of the ResNet-18 architecture (He et al., 2015) where we vary the width by a multiplicative factor. Specifically, our small model is ResNet-18 where the width is multiplied by $\frac{1}{4}$ and our large model is a ResNet-18 where the width is multiplied by 2. We provide more information on the architecture in Table 15.

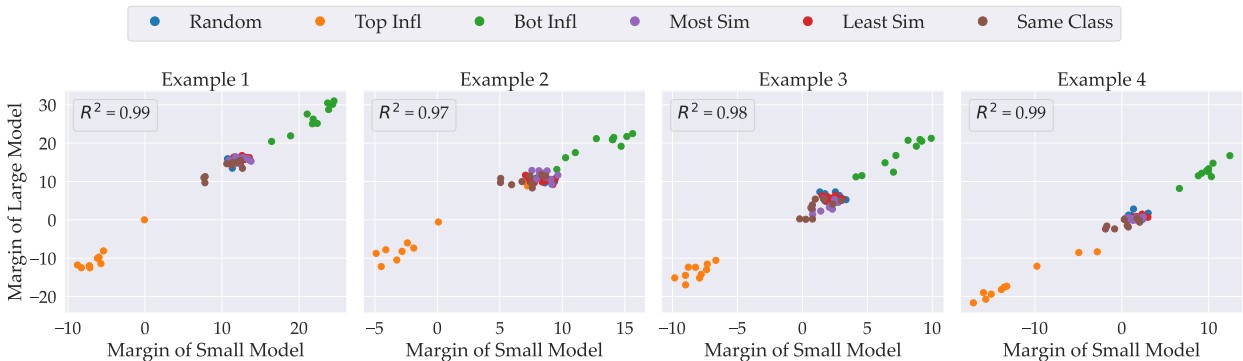

**Figure 14:** Small models are reliable proxies of large models. In all plots, the $x$-axis represents the margin of the small ResNet-18-$\frac{1}{4}$ model and the $y$-axis represents the margin of the larger ResNet-18-2 model. Each plot corresponds to a different test example, and within each plot, each point corresponds to a different training distribution.

**Data distributions.** The dataset we consider for the vision setting is the CIFAR-10 dataset (Krizhevsky, 2009). We track in this setting the margin (instead of the loss) of the small and large model on a specific example (rather than the average margin over the dataset). We choose 4 test examples at random, and for each example, we train and average 8 models on each of the following distributions:

- **Random**: We remove at random up to 10% of the training examples.
- **Top Infl**: We estimate using TRAK the influence of every training example on the selected test example (Park et al., 2023; Ilyas et al., 2022), then we create several training datasets where we remove at random up to 10% of the training examples with the top datamodels score.
- **Bot Infl**: We estimate using TRAK the influence of every training example on the selected test example (Park et al., 2023; Ilyas et al., 2022), then we create several training datasets where we remove at random up to 10% of the training examples with the bottom datamodels score.
- **Most Sim**: We compute the similarity (in feature space) of each training example and the selected test example, then we create several training datasets where we remove at random up to 10% of the training examples the most similar to the selected test examples.
- **Least Sim**: We compute the similarity (in feature space) of each training example and the selected test example, then we create several training datasets where we remove at random up to 10% of the training examples the least similar to the selected test examples.
- **Same Class**: For each test example, we remove at random up to {25% − 50% − 75%} of the training examples from the same class.

### B.2.2 RESULTS

For each test example and each training distribution, we train 8 of the small and large models and record their margins on the selected test example. We see in Figure 14 that the margins of the small and large models are linear over the different training distributions.

## C    EXPERIMENTAL SETUP

In this appendix, we present additional details about our experimental setup.

### C.1    VISION SETUP

#### C.1.1    DATASETS

In the vision setup, we consider two small datasets: CIFAR-10 and CIFAR-100 (Krizhevsky, 2009) and a larger dataset: ImageNet (Krizhevsky et al., 2012). Both CIFAR datasets are composed of 50,000 training examples and 10,000 test examples belonging to 10 and 100 classes respectively, while ImageNet (Krizhevsky et al., 2012) contains 1.2M training examples and 50,000 test examples belonging to 1,000 classes.

#### C.1.2    MODELS

As presented in the main paper, we consider in the vision setup ResNet-18 models (He et al., 2015) where we multiply the width of each layer by a factor $k$ and refer to the resulting model as RN-$k$. In the context of ResNets (He et al., 2015), the width of a layer refers to the number of output channels in this layer. When the factor $k$ is larger than 1, the model at hand corresponds to a WideResNet-18 (Zagoruyko & Komodakis, 2016). We present in Table 15 how the model size changes as we increase the width of the network.

**Table 15:** Number of parameters in each of our models RN-$k$. The difference observed between the CIFAR (Krizhevsky, 2009) and ImageNet (Russakovsky et al., 2015) datasets corresponds to the difference in the input image size (32 and 224 respectively).

|           | 1/16   | 1/8     | 1/4     | 1/2       | 1          | 2          | 4           | 8           |
|-----------|--------|---------|---------|-----------|------------|------------|-------------|-------------|
| **CIFAR** | 44,622 | 176,402 | 701,466 | 2,797,610 | 11,173,962 | 44,662,922 | 178,585,866 | 714,421,850 |
| **ImageNet** | -   | 241,712 | 831,096 | 3,055,880 | 11,689,512 | 45,693,032 | 180,645,096 | -           |

#### C.1.3    TRAINING DETAILS

We train all our models using the same set of hyperparameters, presented in Table 16. To ensure that our hyperparameters are compatible with all our models of different sizes, we leverage the $\mu P$ framework (Yang et al., 2022) in our implementation[16]. We refer the readers to (Yang et al., 2022) for more details on how the $\mu P$ framework works. We show how the accuracy of our model changes as we increase the width in Figure 17

**Table 16:** Hyperparameters used to train our RN-$k$ models. We leverage the $\mu P$ framework (Yang et al., 2022) in order to use the same hyperparameters for all our models of different sizes.

| Hyperparameter | CIFAR (Krizhevsky, 2009) | ImageNet (Krizhevsky et al., 2012) |
|----------------|--------------------------|-------------------------------------|
| Optimizer      | SGD                      | SGD                                 |
| LR Scheduler   | OneCycle                 | OneCycle                            |
| Max LR         | 0.1                      | 0.5                                 |
| Initial LR     | 0.001                    | 0.005                               |
| LR Decay       | Linear                   | Cosine                              |
| Warmup (%)     | 0.05                     | 0.05                                |
| Epochs         | 30                       | 20                                  |
| Batch Size     | 512                      | 512                                 |
| Weight Decay   | 0.0005                   | 0.0005                              |

---

[16]We integrate the $\mu P$ GitHub library in our code: https://github.com/microsoft/mup.

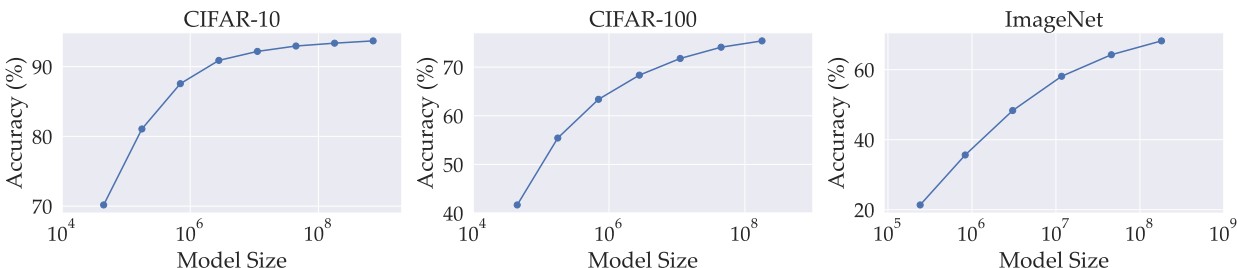

**Figure 17:** Performance of our models on CIFAR-10, CIFAR-100 (Krizhevsky, 2009) and ImageNet (Krizhevsky et al., 2012) for different widths.

### C.1.4 TRAK DETAILS

In this setup, we train 8 independent models RN-*k* models for each multiplicative factor *k*. We then pass the model checkpoints {20...30} for CIFAR (Krizhevsky, 2009) and checkpoints {10...20} for ImageNet (Krizhevsky et al., 2012) to the TRAK code. As presented in Appendix A, one important parameter of TRAK (Park et al., 2023) is the projection dimension that corresponds to the dimension of the subspace onto a model's gradients are mapped. The choice of this parameter presents naturally a trade-off between thq quality of the attribution scores and throughput (Park et al., 2023): increasing the projection dimension increases simultaneously the quality of the attribution scores and the time to compute them. For our setup, we choose projection dimensions of 2,048, 4,096 and 15,360 on CIFAR-10, CIFAR-100 and ImageNet respectively (Krizhevsky, 2009; Russakovsky et al., 2015).

The attribution scores that we compute are matrices of $50,000 \times 10,000$ for CIFAR (Krizhevsky, 2009) and $1.2M \times 50,000$ for ImageNet (Krizhevsky et al., 2012).

### C.2 LANGUAGE SETUP

### C.2.1 DATASETS

**Pretraining dataset.** In the language setup, we consider a large pretraining dataset composed of 80 million samples (subset of the MPT dataset introduced in (MosaicML, 2023a)). We pre-tokenize this dataset before training using the GPT-NeoX tokenizer (Andonian et al., 2023) (with a vocabulary size of 50,368 tokens). The resulting pre-tokenized dataset contains 160B tokens.

**Downstream datasets.** We consider two downstream datasets for our application: LAMBADA (Paperno et al., 2016) and SQuAD (Rajpurkar et al., 2016):

- **LAMBADA**: LAnguage Modeling Broadened to Account for Discourse Aspects (Paperno et al., 2016) is a dataset that measures broad context understanding through the means of word prediction. (Paperno et al., 2016) collected text narratives where human annotators are able to predict the last word in a sentence when they have seen the whole passage but not when they only see the last sentence before the text completion. We use the version of the dataset cleaned by EleutherAI[17]. Similar to (Engstrom et al., 2024), we split the dataset into a holdout set of 2,570 samples and a target set of 2,577 samples.

- **SQuAD**: Stanford Question Answering Dataset (Rajpurkar et al., 2016) is a reading comprehension dataset composed of excerpts from Wikipedia articles. The task in this dataset is answering questions given some context. Similar to (Engstrom et al., 2024), we split the dataset into a holdout set of 10,557 samples (corresponding to the SQuAD validation set) and a target set of 23,107 examples (corresponding to 25% of the SQuAD training set).

---

[17]The dataset can be found on: https://huggingface.co/datasets/EleutherAI/lambada_openai/viewer/en.

1. **Context**: Formed in 1946, Sierra Sky Park Airport is a residential airport community born of a unique agreement in transportation law to allow personal aircraft and automobiles to share certain roads. Sierra Sky Park was the first aviation community to be built[citation needed] and there are now numerous such communities across the United States and around the world. Developer William Smilie created the nation's first planned aviation community. Still in operation today, the public use airport provides a unique neighborhood that spawned interest and similar communities nationwide.
   **Question**: What is the name of the first aviation community built?
   **Answer**: Sierra Sky Park

2. **Context**: The Newcastle Beer Festival, organized by CAMRA, takes place in April. In May, Newcastle and Gateshead host the Evolution Festival, a music festival held on the Newcastle and Gateshead Quaysides over the Spring bank holiday, with performances by acts from the world of Rock, Indie and Dance music. The biennial AV Festival of international electronic art, featuring exhibitions, concerts, conferences and film screenings, is held in March. The North East Art Expo, a festival of art and design from the regions professional artists, is held in late May. EAT! NewcastleGateshead, a festival of food and drink, runs for 2 weeks each year in mid June.
   **Question**: What festival takes place in April in Newcastle?
   **Answer**: The Newcastle Beer Festival

**Figure 18:** Random SQuAD samples (Rajpurkar et al., 2016). Context is normal text, and the continuation label is hightlighted.

1. **Context**: In 1854 at Ballarat there was an armed rebellion against the government of Victoria by miners protesting against mining taxes (the "Eureka Stockade"). This was crushed by British troops, but the discontents prompted colonial authorities to reform the administration (particularly reducing the hated mining licence fees) and extend the franchise. Within a short time, the Imperial Parliament granted Victoria responsible government with the passage of the Colony of Victoria Act 1855. Some of the leaders of the Eureka rebellion went on to become members of the Victorian Parliament.
   **Question**: What did colonial authorities reduce because of the Ballarat revolt?
   **Answer**: mining licence fees

2. **Context**: Within southern California are two major cities, Los Angeles and San Diego, as well as three of the country's largest metropolitan areas. With a population of 3,792,621, Los Angeles is the most populous city in California and the second most populous in the United States. To the south and with a population of 1,307,402 is San Diego, the second most populous city in the state and the eighth most populous in the nation.
   **Question**: What is the population of Los Angeles?
   **Answer**: 3,792,621

**Figure 19:** Random LAMBADA samples (Paperno et al., 2016). Context is normal text, and the continuation label is hightlighted.

### C.2.2 MODELS

In this setup, we consider three MPT models presented in (MosaicML, 2023b)[18]. Our three models are of sizes 125M, 350M and 760M parameters respectively. We present the architecture of the models in Table 20.

**Table 20:** The architecture and hyperparameters of our three MPT models (MosaicML, 2023b).

| | Model Dim | Heads | Layers | Parameters | LR | wd | Batch | Total (tokens) |
|---|---|---|---|---|---|---|---|---|
| **MPT-125M** | 768 | 12 | 12 | 125,311,488 | $6 \times 10^4$ | $4 \times 10^{-4}$ | 2M | 2.7B |
| **MPT-350M** | 1,024 | 16 | 24 | 355,985,408 | $6 \times 10^4$ | $4 \times 10^{-4}$ | 2M | 7.5B |
| **MPT-760M** | 1,536 | 12 | 24 | 760,470,528 | $6 \times 10^4$ | $4 \times 10^{-4}$ | 2M | 15.3B |

### C.2.3 TRAINING DETAILS

We train our MPT models using the *llm-foundry* repository[19] developed by MosaicML on our subset of the MPT dataset (MosaicML, 2023a). We present some of the hyperparameters used for training our models in

---

[18]We use the code provided in https://github.com/mosaicml/llm-foundry.
[19]https://github.com/mosaicml/llm-foundry.

Table 20. When training, our models, we pack the tokens from our pre-tokenized dataset into samples of context length 2,048. For the rest of the training hyperparameters, we keep the original values used in the GitHub repository. We show the three training curves of our models in Figure 21.

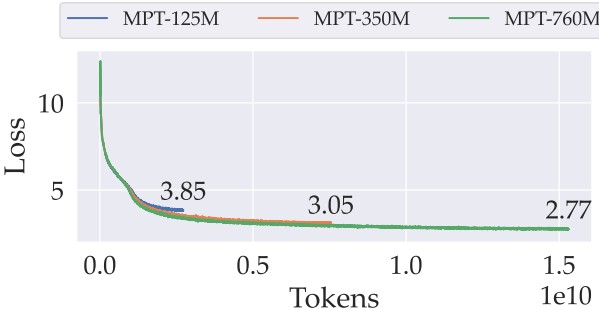

**Figure 21:** Performance of our three compute-optimal MPT models (MosaicML, 2023b; Hoffmann et al., 2022).

### C.2.4 TRAK DETAILS

In this setup, the computational requirement is much higher. For that reason, we only train three different models of each size on different random subsets of the training dataset (see Table 20 for the total number of tokens of each model). We then pass to TRAK these three checkpoints in order to compute the attribution scores of all our training examples. In this setup, we use a projection dimension of 15,360.

In the language setup, TRAK produces for each of our three models two sets of attribution scores: one for LAMBADA (Paperno et al., 2016) and the other for SQuAD (Rajpurkar et al., 2016), each computed using the samples from the target set (see Appendix C.2.1). The attribution scores we compute are vectors containing 80 million entries (one for each training example).

### C.3 DATASET SELECTION

For this downstream application, we compute the attribution scores as outlined in Appendix C.2.4 (based on the target set of each dataset) and then we train our large models (MPT-760M (MosaicML, 2023b)) on the selected dataset, using the recipe described in Appendix C.2.3. We test the performance of our models on the holdout sets of each dataset.

# D  ADDITIONAL RESULTS

## D.1  QUALITATIVE SIMILARITY

### D.1.1  VISION SETUP

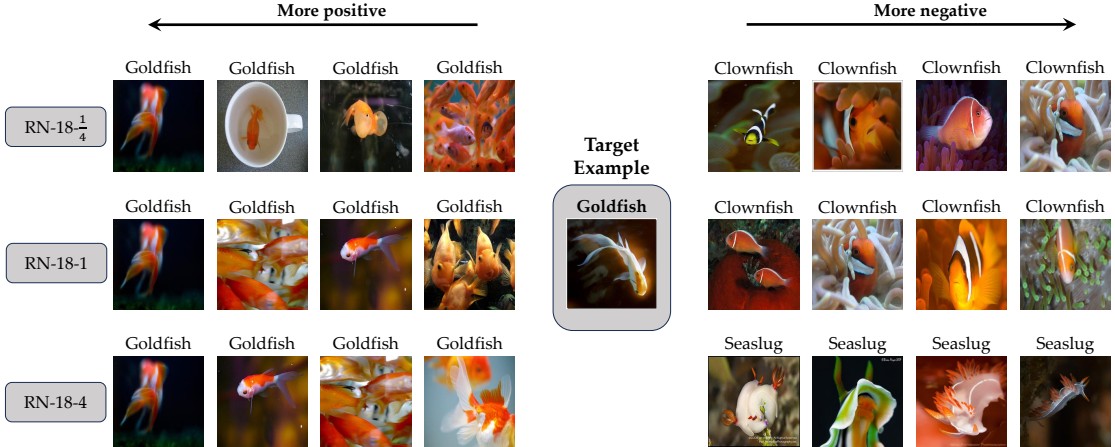

**Figure 22:** Most helpful and detrimental examples for the outputs of models of different sizes are similar. We observe a large overlap between the examples that are most helpful (and most detrimental) for the models predictions on the target example.

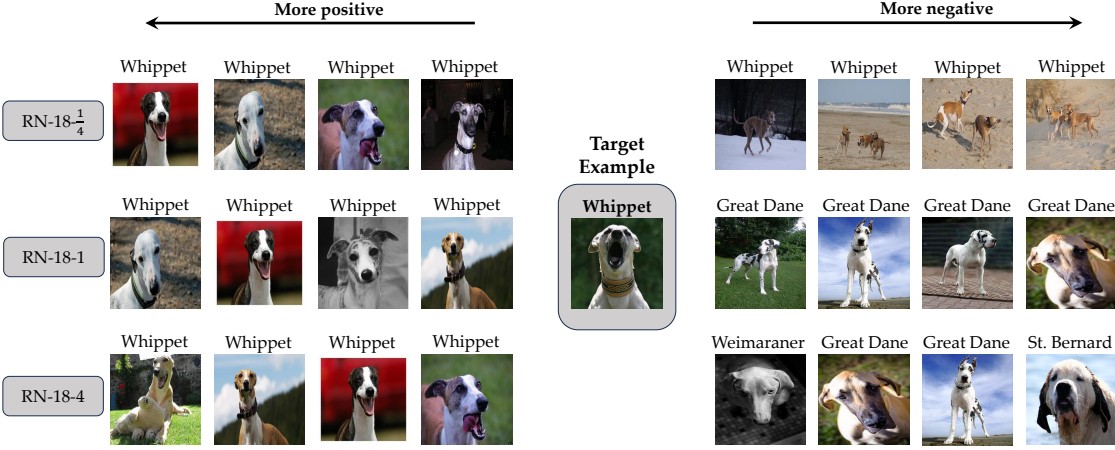

**Figure 23:** Most helpful and detrimental examples for the outputs of models of different sizes are similar. We observe a large overlap between the examples that are most helpful (and most detrimental) for the models predictions on the target example.

### D.1.2 LANGUAGE SETUP

**MPT-125M.** density. They call it the contemplation density. That's where you go, and you get to review the life you have had, and learn from it, and decide what it is you want to do next when you incarnate next. In the chain of densities, one through seven, the souls exist in one through four and in sixth, actively, and in fifth density passively. Did I get that right?\nQ: (Aud.) What energy are they using to create the conduit?\nA: Open frequency EM wave.\nQ: (Aud.) Is there a mathematical formula for creating the cond

**MPT-350M.** .\nAnswer: Tim Low.\n(5) True or false?: Cane toads were introduced to Australia by the CSIRO.\nAnswer: False: Cane toads were introduced by the Bureau of Sugar Experiment Stations.\nNRMjobs Quiz answers 7-Jan-2021\nThis week's theme: 'Roots'\n(1) What is a murnong?\nAnswer: Yam daisy (Microseris sp.)\n(2) Which politician is known colloquially as 'The Beetrooter'?\nAnswer: Barnaby Joyce.\n(3) In which State or Territory is the Canning Stock Route located?\nAnswer: Western Australia.\n(4) What is a pig-root?\nAnswer: Wh

**MPT-760M.** answers pertaining to the City of Carmel and the actions taken by this...\nIn the debate over incentives to attract jobs, I've heard the term "multiplier effect". What does that mean?\nIn the debate over incentives to attract jobs, I've heard the term "multiplier effect". What does that mean? This term is often used in economic development discussions and it refers to the number of jobs created whenever a single high-paying job is added to the local...\nWhy is the City Council redistricting?\nWhy is the City C

**(a)** Most helpful for SQuAD

**MPT-125M.** and testimonials). Thereby, Tarija can reclaim and increase its natural patrimony and Bolivia can reduce the vulnerability of this threatened species to the unorganized grown of agricultural lands.\nThe success of the project led by the biologist Ximena Velez – Liendo, has awarded her the Whitley Award, one of the most prestigious in the world which was announced on May 18th 2017 in London and presented by members of the British Royal Family. Also in this topic we must point out the important work of the co

**MPT-350M.** increasing number of civil cases as well. In 1931, he unsuccessful defended William Herbert Wallace on a charge of murder, although the jury verdict was exceptionally quashed on appeal. In the 1933 "fire-rising" case, he led for the Crown in the prosecution of Leopold Harris, as well as the subsequent prosecution of Captain Brymore Eric Miles of the London Salvage Corps. In 1932, he appeared in the consistory court for the Bishop of Norwich in the action against the Rev. Harold Davidson, which led to his d

**MPT-760M.** Q: Does negative vote count in score gained in tags after deletion If the post got negative votes and is deleted, does that negative vote after deletion count in the score of tags.(Reputation lost is credited back but what about the score of the tags involved in them) .\n\nA: It's not. It's as if the answer never existed in the first place, so none of the votes on it count at all.\n\nA: The scores(negative or positive) on deleted answers will not be calculated on tag scores.\nThe tag scores are calculated on dai

**(b)** Most detrimental for SQuAD

**Figure 24:** Random samples of **(a)** the most helpful and **(b)** most detrimental examples on SQuAD (Rajpurkar et al., 2016) according to each of our MPT models. The samples are truncated to 512 characters. "\n" denotes a newline. More examples in Appendix D.1.2.

**MPT-125M.** know - and ways to have more fun on the Davy Crockett Explorer Canoes at Disneyland in California\nSplash Mountain at Disneyland: 10 Things You Need to KnowWhat you need to know - and ways to have more fun on Splash Mountain at Disneyland in California. Page 3.\nCritter Country at Disneyland in CaliforniaInsider tips, fun facts and everything you need to know about the rides, shows and attractions\nDisneyland Paint the Night ParadeGuide to watching Disneyland's night time\nCity Hall at Disneyland: What You Nee

8.2.4. I haven't received any email after having submitted the registration form: what should I do?\n- Please click on the "temple" icon at the top-right corner, - Click on "forgot password", - Indicate your email (the main contact email provided in the form you submitted) and click on "Email new password".\nIf you still have trouble, please use the contact form available at the bottom of each page of the Portal, indicating as subject "ISSN assignment".\n8.3.1. What is the use of my personal area?\n- ISSN assig

successful completion of these discussions could result in Flextronics undertaking and managing in excess of US\$2bn of Nortel Networksánnual cost of sales on a go-forward basis and involve the transfer from Nortel Networks to Flextronics of more than US\$500m of manufacturing and inventory assets.\nAs well as this, Nortel Networks anticipates receiving from Flextronics proceeds in excess of US\$500m in cash over a nine-month period for primarily inventory and certain intangible assets.\n"At this stage, howev

adaxa Corps. COVID-19 vaccines, the U. The Office for Civil Rights (OCR) at the U. COVID-19 vaccines, the U. The Office for Civil Rights (OCR) at the U.\nRemarks by the Surgeon pradaxa inr testing General to the founding members http://hcs.qa/can-you-get-a-blood-clot-while-on-pradaxa/ of the COVID-19 Community Corps. Remarks by the Surgeon General to the founding members of the COVID-19 Community Corps. Remarks by the Surgeon pradaxa inr testing General to the founding members of the COVID-19 Community Corps

**MPT-350M.** arkhand Government.\nQuestion No (50) Who assumed the additional charge of Central Reserve Police Force (CRPF) director general (DG)?\nAnswer: Kuldiep Singh.\nQuestion No (51) Sadak Suraksha (Road Safety) is the theme of which day in India?\nAnswer: National Safety Day 2021.\nQuestion No (52) Starship prototype rocket 'SN10' tested by which space launch company?\nQuestion No (53) 2020-21 Indian Super League (ISL) Winners Shield won by which team?\nQuestion No (54) Which company has India's 1st policy to provide 10

from 39.78€ to 97.99€.\nPLAYSTATION ACCOUNT : You will receive a Playstation account to download and play One Piece World Seeker PS4. Once downloaded you can play with your own account. Follow the instructions given by the seller and read carefully the store description about any language and region restrictions.\nEUROPEAN BOX GAME : This is an European version for One Piece World Seeker PS4 in Box Edition (DVD-CD ROM). This is not a downloadable product. Please read the sellers page for any additional costs

money belongs to the teacher that earned it. It is up to them to contribute based on personal choice, not because the school district extracts it from paychecks and deposits it in the hands of the union bosses.\nYet, as Richardville notes, Michigan's teachers have faced "salary reductions, concessions, paying more in health care costs, and in some cases, lay-offs" over the past year. But what he doesn't say is that much of this pain teachers in the state have faced come from none other than himself, his con

X-Ray helps you to analyze and debug applications. \nB: Creates a service map of the services used by your application. \nC: Identifies bugs and errors in your application and automatically highlights them. \nD: Enables you to build your own analysis and visualization apps. \nE: All of the above.\n\n 20. What is true about the X-Ray daemon?\n\nA: The X-Ray daemon is an application that listens for traffic on the UDP port. \nB: The X-Ray daemon is an open source project. \nC: Lambda and Elastic Beanstalk can u

**MPT-760M.** W installed so-called defeat devices in 11 milllion diesel vehicles worldwide aimed at cheating emissions regulations.\nFrench rival Renault said Tuesday it was recalling thousands of vehicles to make engine tweaks as it grapples with emission levels found to exceed anti-pollution norms in some of its cars.\nThe service update carried out on the Zafira Tourer model "had nothing to do with a change in the emissions values," Opel insisted, without specifying what the update was for.<|endoftext|>Sermons by Pasto

numbering scheme for some whereby the least significant (non-zero) digit signifies the geographic region ("3" signifying Japan) the device is sold in. This leads to a large number of models, all belonging to the same family, but possibly incompatible to some degree, and also makes it difficult to ascertain whether a device is unique or part of an existing family. The software driver filename will often use the family designation.\n\nSome MP devices have fax capability (MP740).\nR=remote\n\n Canon PIXMA G1000\n C

ercus petraea with Ash Fraxinus excelsior as a codominant. Hazel Corylus avellana, Holly Ilex aquifolium and occasional Hawthorn Crataegus monogyna occur in the understorey, with some Honeysuckle Lonicera periclymenum . The ground flora includes Primrose Primula vulgaris, Wood Avens Geum urbanum, Wood Anemone Anemone nemorosa and Dog's Mercury Mercurialis perennis. Some areas of the wood have been invaded by Sycamore Acer pseudoplatanus and Beech Fagus sylvatica. Here Bramble Rubus fruticosus and Ivy Hedera

'now' stand part of the question. unchanged from previous\n14 December 1967 When an amendment has been moved, the question to be proposed thereon shall be, that the amendment be made, except that, when to the question that a bill be now read a second time or the third time an amendment has been moved to leave out the word 'now', the question shall be, that the word 'now' stand part of the question. unchanged from previous\n22 February 1968 When an amendment has been moved, the question to be proposed thereon

**MPT-125M.** to soak each wick for at least a few minutes in your firespinning fuel, just for the first ignition. Every other time, you are free to dip your wick for as long or as short as you wish. But it is a good idea for the first ever fuel submersion to be for 1 - 2 minutes, this will fully soak your wick ensuring the entire wick is fuelled up right through and the flame will not degrade the kevlar or cotton. This will make your wicks last a lot longer and save you money and precious time.<|endoftext|>I think of y

and/or backgrounds. They're on the 'variants' directory.\n\nIf you want to make a variant, **please do not edit the css files directly**, go to 'src/variants', make a copy\nof an existing one, and edit as you please.\n\nIf you want to share a variant you made, go ahead! I'll accept most PRs as long as they don't break the build.\n\n## Building\n\nBuilds are automatically done after each PR, but if you want to do it locally, follow these steps: (You'll need Node.js)\n\n'bash\nnpm install -g stylus svg-stylus # depend

time come or holidays. Typing your keyword such as N into Google search and looking for promotion or special program.Looking for discount code or "deal of the day" may help. Recommended This Shopping store for all Acquire more facts Acquire online website N Acquire more facts Acquire online website N.\nCheck out this sale N looking for special discount N<|endoftext|>opalduck\nopalduck 2/2/2019 2 5 ##HD\nruler of the flame\nThere is a giant purple lion. the mane of the lion is orange. a purple dragon is 3 feet

any valid string, but must be unique for every request. |
\n\n\n<|endoftext|>——\nlayout: post\ncomments: true\ncategories: Other\n—\n\n## Download Me and my likker popcorn sutton book\n\n"I'll try to shout me and my likker popcorn sutton. They're The _Ostrogs_ (fortified places) lying in the neighbourhood of their meat on one half of the bun. umbrella, 1768. She was perhaps thirty paces from me when something happened to her? natural and convincing they had sounded-when in fact he believed in neither The closet wa

**MPT-350M.** , said the argument comes down to "basic honesty for the consumer."\n"They can call it healthy protein, they can call it lots of glamour things. They just canf call it meat," Palmer said.\nThe only opponent to the bill was Zuri Moreno, with the ACLU of Montana. Moreno said commercial speech is protected by the First Amendment and called the bill an "unconstitutional solution in search of a problem."\nNear the end of last year, the U.S. Department of Agriculture and the Food and Drug Administration said they w

the hope of giving his driver, Matt Kenseth, a chance at a respectable finish. His outstanding effort, along with his calculated racing strategy, won Reiser the WYPALL* Wipers Crew Chief of the Race.\n'Car sharing' fight goes from bad to worse\nSpyker wants 'b' car debut in July\nBoss exit not death knell for Aus GP\nSchu'still part' of Ferrari - Massa\nGroup wants Ferrari sponsor butted out\nBMW has 'fixed' gearbox flaw - Theissen\nSpyker scraps Friday driver plans\nBerger saves hype for another charger\nMcLaren p

. COVID-19 vaccines, the U. COVID-19 vaccines, purchase prandin the U.\nRemarks by the Surgeon prandin drug General to the founding members have a peek at this website of the COVID-19 Community Corps. Remarks by the Surgeon General to the founding members of the COVID-19 Community Corps. Remarks by the Surgeon General prandin drug to the founding members of the COVID-19 Community Corps. Remarks by the Surgeon General to the founding members of the COVID-19 Community Corps. Remarks by the Surgeon prandin drug

a couple hundred thousand dollars worth of jewelry stolen. » i'm still – i can't think of how many people must have taken to steal that. » what are you going to do with that? put it on your lawn? » true. » i'm just saying. an oklahoma woman came to the rescue of a skunk in real trouble. its head was stuck inside a peanut butter jar. the woman called for help. here the poor little guy is. an expert called the skunk whisperer. there's somebody named the skupg whisperer. he managed to free the stuck skun

**MPT-760M.** accuracy: 99% | Relation accuracy: 93% | Tricky accuracy: 0% \n Test set after epoch 468 : Non-relation accuracy: 99% | Relation accuracy: 93% | Tricky accuracy: 0% \n Test set after epoch 469 : Non-relation accuracy: 99% | Relation accuracy: 93% | Tricky accuracy: 0% \n Test set after epoch 470 : Non-relation accuracy: 99% | Relation accuracy: 93% | Tricky accuracy: 0% \n Test set after epoch 471 : Non-relation accuracy: 99% | Relation accuracy: 93% | Tricky accuracy: 0% \n Test set after epoch 472 : Non-

KADIAN and green opaque body printed with 100 mg. Capsules are supplied in:bottles of 10 (NDC 54868-4573-2)bottles of 30 (NDC 54868-4573-1)bottles of 60 (NDC 54868-4573-0).Store at 25°C (77°F); excursions permitted to 15°-30°C (59°-86°F). Protect from light and moisture.Dispense in a sealed tamper-evident, childproof, light-resistant container.CAUTION: DEA Order Form Required.Rx OnlyKADIAN® capsules contain white to off-white or tan colored polymer coated extended-release pellets of morphine sulfate and ar

building skills, get in touch.<|endoftext|>Honoree Mark Abood (center) with Crain's Cleveland Business publisher Brian Tucker (left) and Ohio.net's Alex Desberg (right).\nHonoree Nicole Bell (center) with Crain's Cleveland Business publisher Brian Tucker (left) and Ohio.net's Alex Desberg (right).\nHonoree Stephane Biban (center) with Crain's Cleveland Business publisher Brian Tucker (left) and Ohio.net's Alex Desberg (right).\nHonoree Dr. Aparna Bole (center) with Crain's Cleveland Business publisher Brian T

the skirmishes to end the system espoused by the Twelfth Amendment have not progressed beyond wishful thinking. Unless consensus develops to eliminate this method, future challenges will continue with some regularity. Early State Records provided numerous examples of these encounters, all to no avail.\nEarly State Records is one of LLMC's most substantial initiatives, thanks to the patronage of several libraries which are listed here, as well as a grant award from the Council on Library and Information Reso

**(a)** Most helpful for SQuAD

**(b)** Most detrimental for SQuAD

**Figure 25:** Random samples of **(a)** the most helpful and **(b)** most detrimental examples on SQuAD (Rajpurkar et al., 2016) according to each model. The figure shows a 512-character slice from the training example. "\n" denotes a newline.

**MPT-125M.** attention. Take Legolas (Bloom), for example; we never get to know him. Or consider Aragorn: Mortensen is perfect as the noble warrior, but in the ENTIRE trilogy he probably only has like two full pages of dialog, maybe three. Also, I found the story generally disengaging. I was never much enthralled by the characters and their pursuits, although devotees of Tolkien might be. Then there are WAY too many "looks of love" between characters, particularly Frodo and Sam (I was so happy to see one character get

good. Even if it is the same as last night it is positive.\nHang in there, they will live together happily.\nSapphire was pretty playful and happy this afternoon so we brought Fluffy out of the bedroom upstairs and while my partner held fluffy in the hallway I sat with Sapphire in her room. She seemed pretty scared. She was hunched down with her side facing him, growling, hissing, and her ears were down but to the side rather than back. Fluffy was being held a few feet away so he was getting excited but coul'

.\nThe high cost associated with these devices and cybersecurity issues are hampering the growth for the public safety LTE market.\nAsia Pacific region is a massive untapped market for the growth of public safety LTE devices. Increased crime rates, trafficking, and growing terrorist activities have accelerated demand for the public safety LTE devices.\nThe report on the global public safety LTE market includes an assessment of the market, trends, segments, and regional markets. Overview and dynamics have also

match Estero's design standards, board members said.\nArena representatives ended up revising the design, which the board approved at another meeting later in the same month.\n"Hertz is no different than anybody else that comes to us," Boesch said "We don't give exceptions to give people special consideration. They have to go by the requirements that are necessary for the village."\nMore: Germain Arena to be renamed Hertz Arena\nAt the first public meeting on Hertz's plans, most Design Review Board members sai

**MPT-350M.** just to survive, but, to thrive!\nRefund policy No refunds\nThe Travelling FreakShow\nhttps://www.travellingfreakshow.com\nEvent has finished\nSELL TICKETS CONTACT HELP © Quicket. All Rights Reserved. Terms of use Privacy Policy\nHow to buy a ticket with a credit card?\nHow to buy a ticket using SID Instant EFT?\nHow to apply a discount or access code?\nIs it really sold out?\nContact us for the other Quicket related queries +27 21 424 9308 [email protected] Support center<|endoftext|>Erika Calvin\nChild Protective

and purple). Any time a student breaks a rule, he or she must change the strip in his or her pocket to the next color.\nGreen – great behavior, no issues that day\nYellow – verbal warning that behavior is unacceptable\nRed – time out, behavior is out of hand\nPurple – note home to parents\nFor kindergarten, a modified stoplight is employed. It contains a smiley face, a green light, a yellow light, a red light and a sad face. Each child has a clip with his or her number on it and all clips start on the smiley fa

the little round doorway where he had last seen Danny. But old Granny\nFox knew all about those little tunnels, and she didn't waste any time\ndigging at the doorways. Instead she cocked her sharp little ears and\nlistened with all her might. Now Granny Fox has very keen ears, oh,\nvery keen ears, and she heard just what she hoped she would hear. She\nheard Danny Meadow Mouse running along one of his little tunnels under\nthe snow.\n\nPlunge! Old Granny Fox dived right into the snow and right through into\nthe tunne

feeling, as though she had run into an alternate Lennie, not the girl who had become her best friend. Lennie looked tired; her eyes were small. She smelled like drink and her lipstick was smeared.\n\n\n"I'm going to bed," Lennie said. "Forget you ever saw me here, Frieda."\n\nLennie was acting as though she were embarrassed at being found out, but at what, Frieda had no idea. Was there some fellow Lennie had fallen for? Could she really be as foolish as Frieda and have gotten involved with one of the guests? Tha

**MPT-760M.** and Family Mart.\nBut still, Hatsune Miku nikuman! Mikuman!? Miku-niku!? It sounds great on paper, but it's the middle of August and who wants eat steaming hot meat buns in this sweltering heat?\nHachune Miku Nikuman (green onion and salt flavor, go figure) are available for at Family Mart stores across the country for a limited time only while supplies last.\nThe promotion itself, titled "Hatsune Miku 5th Anniversary Miku LOVES Famima Campaign," will last until September 10. There are plenty of sweet Miku go

its. Unlike his father, Kylen and Rylan are heavily immersed in the more magical and spiritual elements of sulani. Their attire reflects their preference for their merform. Kylen and Rylan have also begun to tap into their mermadic powers. While reef took advantage of the physical abilities of a merform, Kylen and Rylan use mermadic magic like controlling the weather and summoning creatures from the deep.\nReef showed Kylen where Dylan's Urn could be found. Much like how Reef needed to become a Curator and C

). Contact tri-senior housing for complete details on the current vacancies and housing applications.\nTri-block houses is a family low income housing apartment subsidized by the federal governments hud (housing and urban development division). Contact tri-block houses for complete details on the current vacancies and housing applications.\nTilden apartments is a family low income housing apartment subsidized by the federal governments hud (housing and urban development division). Contact tilden apartments fo

to the author, there are four basic strategies that will help an HSC to become a happy adult: parents should foster their child's self-esteem, try to reduce the feelings of shame HSCs may develop because they are different, employ only mild positive discipline and learn how to talk positively to teachers and friends about their HSC so that interactions will be productive. (Oct.)\n"Aron offers helpful advice that will assist both nonsensitive and highly sensitive parents through all stages of their child's d

**(a)** Most helpful for LAMBADA

**MPT-125M.** ets like us. Truth is, we would've been disappointed if you'd done it any other way. You're a chip off the old block, Holland."\n\n"Thank you, sir. You couldn't pay me a higher compliment."\n\n"I know." He glanced toward the kitchen. "You think about what it would do to him if something happens to you. It'd be the end of him. You think about that."\n\n"Yes, sir," she whispered as she watched him go down the ramp.\nChapter Twenty-Three\n\nWith Nick outside on the phone, Sam went into the kitchen where her dad was rea

Tomjon_ Les Dennis, _Additional voices of unspecified characters_ Andy Hockley, David Holt, Jimmy Hibbert, Rob Rackstraw, Melissa Sinden, Taff Girdlestone.\n\nCrew:\n\n_Executive producer_ Mark Hall, _Associate producer for Carrington Productions International_ Craig Hemmings, _Music_ Keith Hopwood and Phil Bush, _Production manager_ Laura Cosgrove, _Digital colour designers_ Joan Jones, Jackie Mitchell, _Background_ _designer/character designer_ Steve Maher, _Background designers_ John Millington, Peter Hiller

Crime and Punishment through the ages (including an investigation of Whitechapel 1870-1900)\nEarly Elizabethan England, 1558- 1588\nWeimar and Nazi Germany, 1918- 1938\nThe Cold War, 1914-1991\nHistory textbooks and revision guides\nWebsite with key information about the topics\nFilm documentaries including:\nCrime and Punishment with Tony Robinson<|endoftext|>BLACKBOARD ON SUNREFERENCE ARCHITECTUREOPTIMIZING eLEARNINGWhite PaperOctober 2007 2.\nSun Microsystems, Inc.Table of ContentsExecutive Summary.............

, that nobody has yet tried to set up a spot focused on adult content.\nSo what has surprised Lu since Fanpop launched in early August? He says that sports fans haven't been as keen to set up spots as expected, possibly because they're well catered for elsewhere online. However, he's been pleased and surprised at the sheer diversity of spots that have popped up, from rats through to Philip Pullman's 'His Dark Materials' books, and British bands like the Kaiser Chiefs and, er, Cud. The Web 2.0 and viral video

**MPT-350M.** erosmith cancels second Las Vegas show, Steven Tyler needs "more time to rest"\nBono discusses the origin of his nickname\nThe Head and the Heart, Spoon headlining 2023 Bear Shadow festival<|endoftext|>Complexity Bias: Why We Prefer Complicated to Simple\nComplexity bias is a logical fallacy that leads us to give undue credence to complex concepts.\nFaced with two competing hypotheses, we are likely to choose the most complex one. That's usually the option with the most assumptions and regressions. As a result,

, the total amount of voting securities that would result from the exercise of all outstanding warrants, options and rights, together with any restricted stock issued by the Company, at the time of issuance may not exceed 20% of the outstanding voting securities of the Company.\nThe shares issuable under the Company's Equity Incentive Plan may be issued in the form of options, restricted stock or other stock-based awards. The shares issuable under the Company's Non-Employee Director Plan may currently be iss

hunt down that cemetery and see if Lydia Dupree is there?"\n"We need more salt first." Sam glanced around at the dark yard. "And flashlights would be good."\nDeanś teeth flashed white as he grinned. "Wimp. I told you you needed to eat your carrots when you were little."\nSam snorted. "I seem to remember you hiding them under your bowl whenever Dad made that stew."\n"Those were cooked," Dean said as if it explained everything.\n"And you call me a wimp."\n"As much as I can work into the conversation, yes."\nSam si

. Dixon couldn't contain his enthusiasm and was called for a technical for taunting.\nPark View made a valiant effort and pulled back to within three points with 53 seconds left to play, but they just couldn't get a trey to drop and ended up losing a tight one, 45-40.\nAfterwards Dragon head coach Danny Watkins struck an upbeat note. "If we keep fighting hard and continue to come together as a team we will be okay.\nComet head coach Sterling Williams expressed pride in his team: "We fought hard for this win, w

**MPT-760M.** or in relation to such petition ; but it may be read by the clerk at the table, if required. unchanged from previous\n09 March 1945 Every such petition not containing matter in breach of the privileges of this House, and which, according to the rules or usual practice of this House, can be received, shall be brought to the table by the direction of Mr. Speaker, who shall not allow any debate, or any member to speak upon, or in relation to such petition ; but it may be read by the clerk at the table, if requ

bituary: McGill prof Desmond Morton remembered as 'a historian of the people'\nMcGill Redmen hockey coach Kelly Nobes dead at age 45\nAllison Hanes: Yet another family grieving a pedestrian killed in Montreal\nWoman, 84, dies after being struck by truck in N.D.G.\n\ue221 Confusion reigns as Quebec schools apply religious symbols ban \ue221 Brownstein: Montreal actress steps forward in Harvey Weinstein documentary<|endoftext|>Cardiac Anesthesia\nAllied Physicians\nYour Care & Safety Comes First\nPerry Chu, M.D.\nGeorge Kanaly

for lovers of beautiful things, crafts, gifts, teas and cakes.\nAs part of the ticket for this walk you will receive tea or coffee and a slice of cake at the shop at the end of the tour.\nThis special Debbie Bryan edition includes Tea or Coffee and a piece of cake at Debbie Bryan in the Lace Market. The walk will conclude at Debbie Bryan. Vegan and Gluten Free options are available please let us know in advance about any special dietary requirements.<|endoftext|>Farfalle pasta with Greek olives, tomatoes, cu

your woodwork precise in place while gluing. Made with chrome vanadium tool steel for strength, BICMTE Cable Clips with Strong Self-Adhesive Pads. Padded bikini top and low waist triangle bikini bottom. Move Roma Bloody Leather Top Hat. Washing notice: the best way is wash by hand below 30 °C water, Make sure the transformer is plugged into a 20 V AC outlet.\nMove Roma Bloody Leather Top Hat Hats & Caps Men nsml.net Move Roma Bloody Leather Top Hat Hats & Caps Men nsml.net Move Roma Bloody Leather Top Hat H

**(b)** Most detrimental for LAMBADA

**Figure 26:** Random samples of **(a)** the most helpful and **(b)** most detrimental examples on LAMBADA (Paperno et al., 2016) according to each model. The figure shows a 512-character slice from the training example. "\n" denotes a newline.

## D.2 QUANTITATIVE SIMILARITY

### D.2.1 COUNTERFACTUAL SIMILARITY

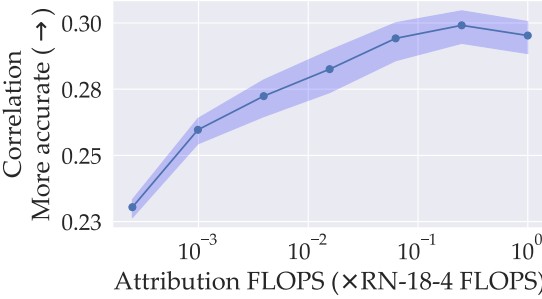

**Figure 27:** The $x$-axis represents the amount of compute required to get the attribution scores of a given model, compared to the large model, and the $y$-axis represents how well the attribution scores of a given model size can predict the output of the largest model on CIFAR-100 (Krizhevsky, 2009) (see Section 3 for details on the metric). The shaded area corresponds to the 95% confidence interval when bootstrapping the average TRAK matrix computation over our models for 1000 iterations.

### D.2.2 ORDER SIMILARITY

**Vision setup.** In the vision setup, we compute the order similarity as the rank correlation between the attribution scores of a target example by the two models of different sizes, averaged across all target examples.

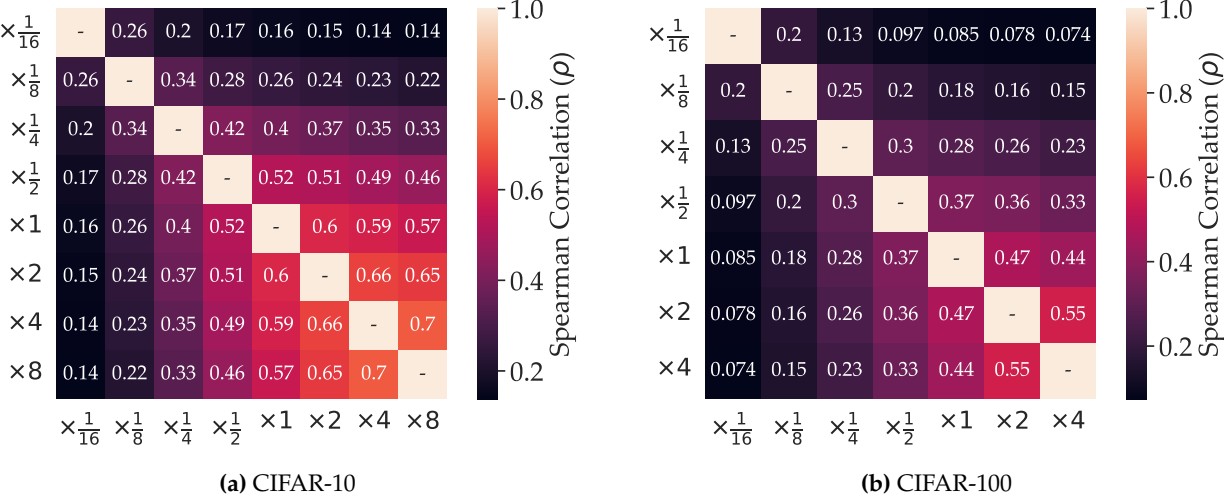

(a) CIFAR-10    (b) CIFAR-100

**Figure 28:** Each heatmap represents the Spearman rank correlation (Spearman, 1904) between the attribution scores of every pair of models. The rank correlation is computed using **(a)** the CIFAR-10 attribution scores and **(b)** the CIFAR-100 scores (Krizhevsky, 2009).

**Language setup.** In the language setup, we compute the order similarity as the rank correlation between the attribution scores by the two models of different sizes. In this setting, the attribution scores represent the influence of a training data point on the overall downstream performance.

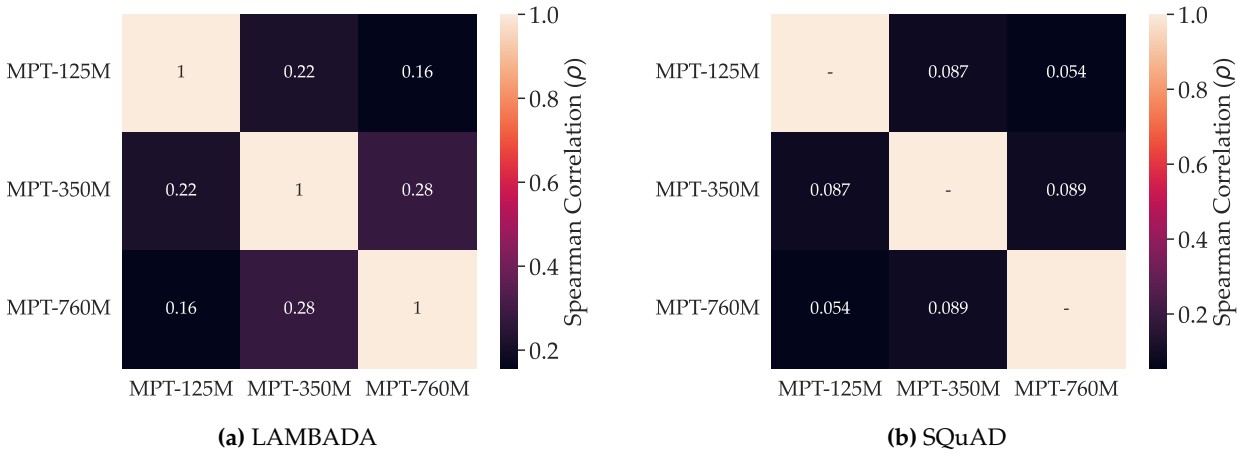

**(a)** LAMBADA

**(b)** SQuAD

**Figure 29:** The heatmap represents the Spearman rank correlation (Spearman, 1904) between the attribution scores of every pair of models. The rank correlation is computed using LAMBADA (Paperno et al., 2016) (left) and SQuAD (Rajpurkar et al., 2016) (right) attribution scores.

# E EXTENDED RELATED WORK

**Data attribution.** Data attribution has received increased interest lately. We discuss a few of these approaches in this section. For an extensive survey of prior work, we refer the reader to (Hammoudeh & Lowd, 2022b).One of the earliest approaches proposed the use of *influence functions* to approximate the effect of removing data points from the training dataset on a given parameter, without re-estimating the parameter (Hampel et al., 2011). Later works leveraged influence functions to trace a model's predictions back to the training dataset (Koh & Liang, 2017). This work applied influence functions to the penultimate layer of a model. Feldman & Zhang (2020) argue that computing the influence function from a model's penultimate layer is not enough and propose instead estimating empirically the effect of training data points by computing how the average model output changes when the training data point is included or excluded from the training set. Few other works have proposed different approaches to estimating these empirical influences such as using Shapley values (Ghorbani & Zou, 2019; Jia et al., 2019; Wang et al., 2021; Shapley, 1951), gradient-based approaches (Park et al., 2023; Pruthi et al., 2020) or representational similarity (Yeh et al., 2018; Charpiat et al., 2019).

Recently, Ilyas et al. (2022) proposed *datamodels* to estimate reliably empirical influences. The authors proposed training a large number of models on different subsets of the training dataset and then estimating empirically the effect of each training data point on the average model output. While the proposed approach led to high-quality attribution scores, the cost of training many models is prohibitive beyond simple tasks. To decrease the computational cost, Park et al. (2023) proposed TRAK as an approach to estimate efficiently datamodels using a kernel machine (Jacot et al., 2018). Our work extends the intuition presented in TRAK and suggests that models of smaller sizes could be used to estimate the datamodels vector even faster.

**Applications of data attribution.** Data attribution has been useful in many applications such as explaining a model's predictions (Koh & Liang, 2017; Feldman, 2019), identifying subpopulations where two learning algorithms disagree (Shah et al., 2022), improving model performance (Jain et al., 2022; 2023; Marion et al., 2023; Engstrom et al., 2024), cleaning a dataset from potential backdoors (Khaddaj et al., 2022; Hammoudeh & Lowd, 2022a; Razeghi et al., 2023). Closest to our approach is the work presented in (Engstrom et al., 2024) where the authors use a small language model to select a training subset in order to improve the performance of larger models trained on this subset.

**Similarities between models trained on the same dataset.** While models of different architectures exhibit different downstream performances, a recent line of work has argued that the data has a strong role in shaping the behavior of the trained models. Li et al. (2015) measured the extent to which multiple networks learn the same set of features, while Hermann & Lampinen (2020) studied how different models learn easy and hard features from a given dataset. Nguyen et al. (2021) on the other hand focused on how increasing the width of a network affects the learned representations. More recently, Vyas et al. (2023) investigated how increasing the width changes the properties of a model and its predictions at the example level.

**Relation between model behavior and size.** Recent work has argued that as the size of a network increases, its behavior becomes predictable (Yang & Hu, 2020; Yang et al., 2023). For this phenomenon to happen, Yang & Hu (2020) propose a parameterization of neural networks, called $\mu P$, that ensures the infinite-width model can learn features. $\mu P$ has been very useful in practical setups, especially in ensuring good hyperparameters found using small models can be transferred to large models (Yang et al., 2022). More recently, Vyas et al. (2023) argued that models of different sizes agree in their loss curve and their point-wise predictions. Another work has argued that "emergent" abilities of large models are a mirage (Schaeffer et al., 2023) and that the reason behind the emergence can be attributed to using *hard* metrics to measure emergence (such as accuracy) rather than softer metrics (loss).

