# OpenReview forum: "Small-to-Large Generalization: Training Data Influences Models Consistently Across Scale"
_ICLR.cc/2025/Conference — ICLR 2025 Poster_

### Official Review · Reviewer_dVoq · 2024-10-28

**Soundness:** 3
**Presentation:** 3
**Contribution:** 2
**Rating:** 3
**Confidence:** 4

**Summary:**

This paper proposes an interesting direction of research in studying how small model performance gives insights to large models. Rather than using existing methods, such as training on small models and trying to extrapolate to large models or influence functions, this paper studies the effect of training data across compute scale. In Figures 1-8, the authors discover that the training performance of a small model generally predicts the performance of a large model quite well, whether the task is NLP, data attribution, or data selection.

**Strengths:**

1. This paper proposes an interesting direction of research in small-to-large model generalization. It is important that we put sanity checks in place when we want to extrapolate large model behavior from small models, and the idea in this paper opens up a new perspective.
2. The experiments carried out in this paper are scientifically sound. The authors show that across tasks ranging from NLP to data attribution and data selection, small models approximate large models pretty well, so long as the model size is reasonable.
3. The presentation in this paper is clear, and details in this paper are easy to find and understand.

**Weaknesses:**

1. While this paper presents an interesting idea, the authors did not present enough results to convince readers why this idea is worth pursuing. From the experiments in this paper, it seems like the conclusion is that "small models approximate large models quite well", period. If that is the case, then this paper acts more as a position paper. If that is not the case, then the authors have not presented enough empirical investigations into the details of small-to-large generalization.
2. Some details in the paper could be better explained with examples and/or discussions. See questions for more details.

**Questions:**

Major problem: The message in this paper is not clear to me.

Context: It seems like the takeaway from this paper can be summarized as "as long as the proxy model is not too small, small-to-large generalization works, which is demonstrated by the results in NLP, data attribution, and data selection".
Questions:
1. Does the takeaway indicate that the problem of small-to-large generalization is solved? With the results in this paper, does it mean that no one has to worry about small-to-large generalization ever again, and we can safely study proxy models without having to worry about sanity checks?
2. Isn't there more to this problem? Does small-to-large generalization work on all data distributions? This paper presents some experiments attempting to answer that question, but what are the rules for small-to-large generalization to work, i.e. what properties does the training data distribution satisfy for this to work?
3. It makes sense that training larger models (parameter size ~ 100B) requires compute that is inaccessible. However, what about quantization? Or training on fewer data points rather than the entire dataset? With the increasingly powerful models we have today, I would expect more interesting findings to arise, i.e. smaller models should not be able to predict large models when the performance of large models gets really good.

In summary, these questions arise because the message in this paper is not clear. To reiterate, this paper brings up an interesting perspective, specifically that "we should apply sanity checks before we attempt to extrapolate proxy models to larger models". However, the materials in this paper does not strength said perspective. (1) If the point of this paper is to prove the perspective, then the authors should look for cases where the proxy model is of sufficient size, but extrapolation fails due to peculiar properties in the training data, compute budget, or any other reason. (2) If the point of this paper is to disprove this perspective, then the authors need to label the situations where it is certain that small-to-large generalization works, and the situations where more sanity checks need to be in place. What the authors seem to be doing in this paper is trying to find experiments that support the transfer from proxy models to large models, which does not bring much insight. In all, I do not believe the experiment results in this paper brings sufficient contribution to the community.

Minor questions to clarify confusion:

1. Do I understand correctly that the experiments in Figures 1-5 are carried out by training the proxy models and large models on one dataset, and testing them on another dataset? And the training dataset is sampled from 1 of 6, the test dataset is sampled from 1 of 4 (bottom of page 2)?
2. Figure 3 presents an experiment finding that is confusing. If the proxy model results are random guessing, how can that extrapolate to large models? Is there an example of that?
3. In Figure 29, shouldn't the correlation be a lot higher?

---

> ### Author Response · Authors · 2024-11-24
>
> We thank the reviewer for their insights. We address below the questions raised.
>
> - *Main takeaway*: In this work we aim to help characterize the conditions under which proxy models are effective; we find that they are broadly effective in the conditions we study at relative compute levels of 10x-100x in NLP and 10^3 in vision, but then slowly drop in reliability after such differences in scale. We do not completely answer the question---an impossibility in an empirically driven field like deep learning---but give evidence that proxy models actually can be effective compared to using the general model even at large differences in scale.
> - *Application to data distributions*: We agree that there is much more to this problem that remains uncharacterized by our work, including properties of the training distribution and properties of the test time distribution. In this work, we focused on popular choices of training/test distributions within the community. We do not claim to fully illuminate this problem and will make the limitations more clear in the revision.
> - *Training 100B models*: As academics, we cannot train 100B models (and quantization here does not move the needle as we are not performing inference; we are training large scale models). We try to provide insights into this setting by performing experiments in small-scale settings that mimic the phenomena seen at large-scale. As an example, see Figure 3: we find proxy models that perform as well as guessing randomly on tasks are effective for larger scale models that predict nontrivially on these tasks (i.e., larger-scale models that have passed the “emergence” compute threshold for these tasks). One can see this as evidence that at large-scale, similar phenomena could arise where smaller proxy models could also predict emergent behavior of models trained with more compute.
> - *Training and test distributions*: We train a model on a distribution and test on another one. Our training distributions are in the legend of Figure 2. Our test distributions are the titles of each subplot of Figure 2.
> - *Results of Figure 3*: We agree it is a curious finding. The proxy models do indeed perform as well as random guessing but are still effective as proxy models for the large-scale models. One example is on the COPA baseline, where MPT-40M models perform worse than randomly selecting but are still effective as proxy models on the large-scale models. We will modify the text to include this example inline, thank you for the insight.
> - *Results of Figure 29*: While the correlation itself is not high, it is the case that the most helpful and most detrimental examples are similar for both big and small models, yet the overall Spearman correlation is weak (as *Reviewer RKkD* pointed out). We conducted a quick experiment in the language modeling setting to validate this hypothesis. By taking the datamodels of our MPT-125M and MPT-760M, we found that the number of samples that are in the top 10% of both datamodels is double the number of samples that are simultaneously in the range [20%-30%]-[30%-40%], …, [80%-90%] of both datamodels. This is also true for the samples that are simultaneously in the bottom 10% of each of the two sets of datamodels.
> We also computed the correlation between datamodels for LAMBADA and found that the spearman correlation is in the range of 20%, much higher than the range observed for SQuAD.

---

> > ### Comment · Reviewer_dVoq · 2024-11-25
> > **I will increase my score**
> >
> > There are still important limitations to this work (see my reviews about the paper), but after reading other reviews and author responses, I am willing to increase my score, because it seems like other reviewers who are more well-versed with the literature believe this paper is an important contribution. I will increase my score to 6.

---

### Official Review · Reviewer_PHtT · 2024-11-02

**Soundness:** 2
**Presentation:** 3
**Contribution:** 3
**Rating:** 6
**Confidence:** 2

**Summary:**

This study demonstrates high correlations between loss predictions in smaller and larger models across various downstream tasks, showing that these correlations hold across different pretraining datasets, downstream datasets, and model scales—up to a certain point. Specifically, this work tests this correlation across two pretraining datasets (Pile and C4) and four downstream datasets (SQuAD, HellaSwag, LAMBADA, and TriviaQA) using academic models, with the largest model at 760M parameters and proxy models down to 56M. They find strong correlations for all datasets except SQuAD, where the correlation is moderate. The study also explores proxy model applications for (i) **data attribution**, showing that the LDS (loss difference score) remains similar even with reduced proxy model sizes, with a caveat that the  LDS values are low, and (ii) **data selection**, demonstrating performance gains in specific downstream tasks when using proxy models, even very small ones, compared to the original model.

**Surprising Findings that I Liked**:

- Loss predictions correlate strongly between models of different sizes, even when accuracy does not (with small models showing random performance).
- This correlation is not just an average across datasets but also holds across samplewise performance distributions showing a peaky distribution near 1.

**Strengths:**

- S1. **Robust Correlations**: The paper demonstrates a high correlation in loss predictions across multiple datasets and model sizes, with findings that hold consistently across various setups.
- S2. **Detailed Samplewise Distributions**: I liked that the paper went beyond just relying on average performance and examined samplewise distributions of performance – showing that it is not some distributional quirk.
- S3. **Proxy Model Applications**: I liked exploration of proxy models in practical applications, such as data attribution and selection. The finding that we can use proxy models to further optimize training efficiency while maintaining a stable LDS score/enable faster data selection is exciting and can lead to impactful followups.
- S4. **Easy to Understand for an Outsider**: The paper effectively motivated the setting for readers unfamiliar with the topic. The proposed method and experiments were relatively easy to understand despite the denseness of the paper.

**Weaknesses:**

I have a few concerns asked in Questions section.

Overall, this work offers promising insights and evidence for the use of proxy models in downstream applications, though I have some reservations. I remain cautiously optimistic and look forward to further discussions with the authors and other reviewers.

**Questions:**

See Questions in the order of importance. If there is lack of time, please prioritize the earlier questions:

Q1. **Control Comparisons for Figure 1**: The divergence between loss and accuracy correlations raises questions about the significance of the observed loss correlations. The lack of control comparisons for loss metrics makes it difficult to assess the meaningfulness of these correlations.
- **Requested Experiment**: Can the paper show $R^2$ correlation comparisons between a random large model and all of the small proxy models trained on different training data distributions. Does this graph still show a high correlation? Similarly, a corresponding comparison between a random large model and small models trained on different training data distributions would be helpful to ground results of Figure 2 and Figure 5.

Q2. **Interpretation of Loss vs. Accuracy Correlation**: The discrepancy between loss and accuracy correlations raises important questions about what the loss metric measures and its utility for downstream tasks, especially given that it does not align with accuracy.
- **Suggested Discussion**: Could the authors explore this point further, clarifying whether loss correlations signify meaningful properties that contribute to downstream performance?

Q3. **Applicability to Models with Higher LDS Scores**: While the LDS score remains stable across smaller proxy models, it is unclear if this stability would persist with higher LDS values, where making proxy models might show far higher performance divergence.
- **Requested Experiment**: If possible, could the authors atleast experiment on CIFAR10 with models with substantially higher LDS scores? (If need be: Using Trak can save lot of compute and time). This would strengthen the findings and alleviate my concerns.

Q4. **Variance in Figure 8**: Including error bars in Figure 8 would help clarify the significance of the observed performance improvements.
- **Requested Experiment**: How do accuracy outcomes vary when training on different, randomly selected data subsets? This would provide insight into the variability and reliability of the performance gains.

---

> ### Author Response · Authors · 2024-11-24
>
> We thank the reviewer for their insights. We address below the questions raised.
>
> - *Correlation with a random model*: The correlation between the output of a random model and any of our models is 0.
> - *Loss vs downstream performance*: We thank the author for their remark. Indeed, loss and performance are not interchangeable. However, they are correlated: lower loss correlates well with better performance. One useful framework to think about the relation is the one presented in [1]. Specifically, the accuracy is some step function of loss: as loss decreases, up to a certain point, the accuracy is close to random, and then after some threshold, the model suddenly has a good accuracy. The loss is such a smoother measure of performance than accuracy.
> - *LDS vs model performance*: We thank the reviewer for their suggestion. We would like to point out that the LDS is independent of model performance. LDS measures how well some datamodels estimates correlate with the actual model output. In practice, the LDS values we get are usually independent of the accuracy of the models we consider. For example, using our exact same experimental setup in the paper, we find that the correlation between the datamodels of our smallest CIFAR-10 model and the predictions of that small model is 0.24 (i.e., LDS=0.24). This is very close to the correlation between the datamodels of our largest CIFAR-10 models and the predictions of that large model (LDS=0.23 – see Figure 6b).
> - *Training on different random subsets*: We ran a quick experiment and the accuracy did not change much by considering different random subsets (+- 0.5%). We would be happy to run more experiments and add error bars to our plot for the camera-ready version.
>
> [1] Are Emergent Abilities of Large Language Models a Mirage? Schaeffer et al. 2023.

---

> > ### Comment · Reviewer_PHtT · 2024-11-25
> > **Thanks**
> >
> > The rebuttal addressed my concerns, I will keep my positive score.

---

### Official Review · Reviewer_RKkD · 2024-11-02

**Soundness:** 3
**Presentation:** 4
**Contribution:** 2
**Rating:** 6
**Confidence:** 3

**Summary:**

The selection and attribution of data is an important ingredient in the training of machine learning models. In practice, it is often intractable to test various data selection strategies multiple times due to the high training costs associated with large models. This work investigates the extent to which a smaller proxy model can be used for data selection and attribution. The authors find that while the behaviors of small and large models do not completely align, they exhibit a correlation in many cases, though this correlation can be weak in specific instances. They support their claim by extensive experiments across several NLP and computer vision tasks.

**Strengths:**

- The problem addressed in this work—using a small model to select training data for a large model—is highly significant for both academia and industry, with substantial potential for saving time and computational resources in the development of large-scale models;
- The paper is very well written and is easy-to-follow;
- The study is extensive, covering conventional strategies in training data construction, including (a) dataset selection and (b) data attribution, as well as tasks across different modalities such as images and text;
- The authors identify several failure cases where the proxy model becomes unreliable. I particularly appreciate Figure 2, which clearly illustrates the cut-off point beyond which predicting the behavior of the reference model using the proxy model becomes infeasible.

**Weaknesses:**

-  Despite the interesting findings and detailed analysis, the models studied in this work may (in my opinion) not be considered "large." For instance, the largest language model examined is below 1B parameters, being significantly smaller than modern LLMs, which often exceed 7B parameters. This raises questions about the analysis and conclusions can generalize to larger models commonly used in real-world scenarios;
- The observation that the correlation between the reference and proxy models varies with scale and specific tasks are expected and unsurprising. The paper lacks insights into which type of tasks are more likely to result in weak correlations, making it still unclear when the proxy model can/cannot be used to predict the reference model;
-  Most experiments focus on classification-like tasks (e.g., multiple-choice answering, final token prediction, image classification) rather than generative tasks (e.g., generating a full sequence of tokens). I understand this is a research focus, but this still limits the significance and impact of the work;
-  Some experimental results do not seem to fully support the authors' claims. For example, in the data attribution experiment presented in Section 3.1, the quantitative correlation (as computed by Spearman’s rho) is low—below 0.30 for all proxy models considered. Note that a correlation of 0.30 is typically not regarded as strong in statistical analysis, this makes it really difficult to say that "the behaviors of the proxy model and the reference model are similar". However, I do think the finding here itself is a noteworthy contribution and already has some implications (see the questions below). Perhaps the authors just need to revise their statement.

**Questions:**

- The results in Section 3.1 indicate that the most helpful and most detrimental examples are similar for both big and small models, yet the overall Spearman correlation is weak. Does this suggest that while the models behave similarly for the extreme samples, they differ significantly for middle samples? A more detailed analysis of this could enhance the paper.
- In the evaluation of NLP tasks, are the language models trained from scratch, or are they fine-tuned? Clarification on this point would be helpful.
- How are the results in Figure 1 (and other figures) collected? Is each point in the figure an average of multiple runs?

---

> ### Author Response · Authors · 2024-11-24
>
> We thank the reviewer for their insights. We address below the questions raised.
> - *Scale of experiments*: As an academic lab, we considered the largest setup that our compute capacity allows; estimating the datamodels for 760M parameter models required 3 weeks on ~50x A100. Experimenting with a larger scale is unfortunately beyond our compute capacity. That said, the compute difference in our setup (up to ~400x in language and 10^5 in vision) is similar to the difference we should expect to see in the real world (e.g., 1B vs 400B params).
> - *Weak correlation (LDS)*: See [GC1].
> - *Results of Section 3.1*: Thanks for a great suggestion. That is indeed the case. We conducted a quick experiment in the language modeling setting to validate this hypothesis. By taking the datamodels of our MPT-125M and MPT-760M, we found that the number of samples that are in the top 10% of both datamodels is double the number of samples that are simultaneously in the range [20%-30%]-[30%-40%], …, [80%-90%] of both datamodels. This is also true for the samples that are simultaneously in the bottom 10% of each of the two sets of datamodels. We would be happy to include a more thorough analysis in the camera-ready version of the paper.
> - *Evaluation of NLP models*: The models are trained from scratch.
> - *Number of runs for Figure 1*: The results in Figure 1 are the result of a single run. We have experimented slightly with increasing the number of runs for the smaller models and that did indeed help the correlation. We haven’t, however, applied this finding due to potential compute costs of pre-training more models.

---

> > ### Comment · Reviewer_RKkD · 2024-11-25
> >
> > I thank the reviewer for their responses. I acknowledge the rebuttal. I will keep my already positive score.
> >
> > My final suggestion is to expand the analysis in Section 3.1 a bit in your final version. Overall I think this is a quite fine contribution. Wish you all the best in the submission!

---

### Official Review · Reviewer_dZhR · 2024-11-03

**Soundness:** 3
**Presentation:** 4
**Contribution:** 3
**Rating:** 6
**Confidence:** 3

**Summary:**

This paper studies how the choice of training data affects model behavior across different computation scales. Specifically, the paper focuses on proxy models: models that are smaller and worse-performing than a reference model that we ultimately care about, but small enough that we'd hope our analyses on the smaller models would carry over to the larger model. The paper makes the following contributions:

- In the first set of experiments, the paper trains models on 10 different training data distributions across 175x compute scales and evaluates them on 6 different hold-out sets. They find a strong correlation between results for small and large models.
- In the first application, the paper studies data attribution, and shows that estimates of training example influence from small proxy models are correlated with the results for the larger reference model.
- In the second application, the paper studies dataset selection, and shows that small proxy models can be used to select subsets of training data that improve the performance of larger reference models.

Overall, this paper concludes that proxy models are effective for predicting how changes in training data will affect larger models.

**Strengths:**

I thought the strengths of the papers were as follows:
- Research question: Studying the effectiveness of proxy models is an important research question. As ML models (e.g. LLMs) get larger, it is infeasible to make all training decisions based on only an enormous reference model. Any progress on knowing whether and when proxy models are useful is important progress.
- Experimental results for data influence across scale: The first set of experiments (Section 2) were quite interesting. The experimental design was straightforward and the results clear to interpret. The fact that proxy models are effective regardless of accuracy was quite interesting to me.
- Clarity: The paper writing was high-quality, and the key conceptual ideas were clearly discussed.

**Weaknesses:**

I thought the biggest weakness of the paper involved the data attribution experiment:
- Weak correlation: The LDS correlation scores are quite low across model scales, never reaching above 0.21 for IMAGENET of 0.22 for CIFAR-10. The paper acknowledges this, but I don't think this is strong enough evidence to support the conclusion that proxy models are useful for this task. It's true that even the correlation score for the reference model itself is quite low, but then this metric seems less useful and not strong enough evidence for the conclusions.
- Metrics: Given the above, it's not clear to me that LDS is the most useful metric. What's the correlation of datamodel weights across small and large models? That seems more closely related to the paper's motivation.
- Estimation methods: The only estimation method that's used is TRAK, so we only see evidence for how predictable TRAK is from smaller proxy models. It would be interesting to see if other influence function-based methods follow the same patterns.
- Confusing axes on Figure 6: The Y-axes in Figure 6a and 6b are labelled differently, but the caption makes it sound like they're the same. Are they the same? And if they're different, what does the Y-axis of 6b mean?

I also thought there were details about the experimental setup that were missing or unmotivated in the main text, such as:
- How many different sizes of small proxy models are used in the experiments in Section 2 and 3? Is it 6 (as Figure 2 would imply)?
- Why is only 0-shot considered for the LAMBADA experiment in Section 3.2, and only 1-5 shots considered for the analogous SQuAD experiment? How do results look for 0-shot on SQuAD and 1-5 shots on LAMBADA?
- Why is the data source selection for Section 2 only on SQuAD? In light of this, how should we interpret the fact that the R^2 for SQuAD is by far the lowest of the test sets?
- Why use different model scales for ImageNet and CIFAR-10 in the data attribution experiment (in reference to: "the largest models are 10^4 times wider than the smaller for ImageNet and 10^5 times for CIFAR-10")?

**Questions:**

Low LDS correlation scores:
- How do LDS scores around 0.2 provide sufficient evidence for proxy model effectiveness?
- Have you considered other metrics (e.g. correlation of datamodel weights)? Or other estimation methods beyond TRAK?

Clarification of experimental details:
- How do results look for 0-shot on SQuAD and 1-5 shots on LAMBADA for the training data selection experiment in Section 3.2?
- Should the y-axis labels in Figure 6a and 6b be the same? If not, what is the interpretation of the y-axis on Figure 6b?
- See additional questions about experimental setup above.

---

> ### Author Response · Authors · 2024-11-24
>
> We thank the reviewer for their insights. We address below the questions raised.
>
> - *Weak correlation*: see [GC1].
> - *Metrics*: Thank you for your suggestion. We have computed the correlation between datamodels weights in Appendix D.2.2. The motivation behind the LDS metric is that we are essentially computing how well datamodels from small models predict the actual output of large models, which is ultimately what we care about.
> - *Estimation methods*: Thank you for your suggestion. Our choice of TRAK was based on its very competitive performance (at the time of the experiment) and its relatively cheaper compute cost. We believe that other influence-based methods would follow similar trends and that is indeed a very interesting avenue for future work.
> - *Labels of Figure 6*: We apologize for the confusion. We had different formats for the same plot and missed unifying the format when submitting. The plots are indeed the same, with 6a representing the ImageNet results and 6b the CIFAR-10 results.
> - *Sizes of proxy models*: Yes, indeed, we tried a range of sizes. We sweep over 6 different sizes for Figure 2. For Section 3, we indicate the sizes we consider are presented in Table 15.
> - *Choice of zero/few-shot for LAMBADA and SQuAD*: In practice, LAMBADA is typically evaluated with 0-shots (as it does not have instructions so there is no need to prompt an pre-trained model to perform a task in a given format), while SQuAD is evaluated with multiple shots (as it does have instructions, so examples are needed to show a pre-trained model examples of the task). As an example, the Mosaic ML Eval Gauntlet uses 0 shots and 3 shots for LAMBADA and SQuAD respectively. Given how closely the results for 1-shot, 3-shot and 5-shot SQuAD match (in relative terms, at least), however, we suspect that the qualitative effect of shots is relatively small.
> - *Choice of selection dataset*: We only selected data sources for SQuAD in Section 2 due to compute constraints; we wanted to include data source distributions selected by “active” dataset selection methods, but would have to reduce the scale of our experiments if we included more “target” distributions (e.g., by selecting for LAMBADA or another test set).
> - *Model scales for ImageNet and CIFAR*: As ImageNet is a much larger dataset, the cost of estimating the datamodels is significantly larger. As such, our largest model on ImageNet is smaller than our largest model on CIFAR-10.

---

> > ### Comment · Reviewer_dZhR · 2024-11-25
> >
> > Thank you for the clarifying response. I'll maintain my original (positive) assessment of the paper.

---

### Author Response · Authors · 2024-11-24

We thank the reviewers for their feedback. We address below some common points and then address each reviewer’s specific concerns separately.

**[GC1]:** The reviewers have expressed worry about the low LDS score (~0.2-0.3) in Section 3. We agree that this is indeed not a high LDS, however, prior work has shown that this LDS can be significantly increased by 1) using more models when estimating the datamodels [1, 2] and 2) modifying one of the parameters of TRAK [2]. Indeed, [2] shows that the LDS can be increased from 0.2 to 0.5 by increasing the number of models for TRAK from 100 to 1,000. Even within our setting, we tried replicating Figure 6b with half the models that are currently used in Fig 6b, and the LDS decreased from [0.2-0.3] to [0.1-0.2]. Increasing the LDS requires increasing the compute required for computing the datamodels, which would make our experiments intractable.

[1] Datamodels: Predicting Predictions from Training Data. Ilyas et al. 2022.

[2] TRAK: Attributing Model Behavior at Scale. Park et al. 2023.

---

### Meta-Review · Area_Chair_7cnq · 2024-12-20

**Metareview:**

The paper undertakes an experimental exploration of how well data selection strategies generalise from small models to larger models. This is accomplished in a method-agnostic manner, by training a large number of models on different data subsets to and measuring correlations across scales. The findings provide some fine-grained insights into when one can extrapolate performance from one scale to another.

The reviewers were generally very positive about the clarity of the paper and significance of the research direction, and were unanimous in their opinion to accept the paper.

**Additional Comments On Reviewer Discussion:**

After the discussion period, the reviewers converged towards a unanimous opinion that the paper should be accepted.

---

### Decision · Program_Chairs · 2025-01-22

Accept (Poster)